# Bioorthogonal photocatalytic proximity labeling in primary living samples

Ziqi Liu [1,2,3], Fuhu Guo[1,3], Yufan Zhu[1], Shengnan Qin[1], Yuchen Hou[1], Haotian Guo[1], Feng Lin[2], Peng R. Chen [1,2] ✉ & Xinyuan Fan [1,2] ✉

In situ profiling of subcellular proteomics in primary living systems, such as native tissues or clinic samples, is crucial for understanding life processes and diseases, yet challenging due to methodological obstacles. Here we report CAT-S, a bioorthogonal photocatalytic chemistry-enabled proximity labeling method, that expands proximity labeling to a wide range of primary living samples for in situ profiling of mitochondrial proteomes. Powered by our thioQM labeling warhead development and targeted bioorthogonal photocatalytic chemistry, CAT-S enables the labeling of mitochondrial proteins in living cells with high efficiency and specificity. We apply CAT-S to diverse cell cultures, dissociated mouse tissues as well as primary T cells from human blood, portraying the native-state mitochondrial proteomic characteristics, and unveiled hidden mitochondrial proteins (PTPN1, SLC35A4 uORF, and TRABD). Furthermore, CAT-S allows quantification of proteomic perturbations on dysfunctional tissues, exampled by diabetic mouse kidneys, revealing the alterations of lipid metabolism that may drive disease progression. Given the advantages of non-genetic operation, generality, and spatiotemporal resolution, CAT-S may open exciting avenues for subcellular proteomic investigations of primary samples that are otherwise inaccessible.

Biological processes are rigorously regulated by proteomic networks organized within subcellular compartments. Perturbations of subcellular proteome networks are linked to various cell abnormalities and diseases, such as tumorigenesis[1] and neurodegeneration[2] that are driven by mitochondrial dysfunctions. Given the high complexity, dynamicity, and fragility of subcellular proteomes, technologies that directly capture subcellular proteomic information in living cells are thus in high demand, which would significantly facilitate biological discovery as well as therapeutic development[3]. In recent years, the biocatalytic proximity labeling approaches (e.g., BioID, APEX)[4,5] have emerged as a milestone for in situ biological study by enabling on-demand capture of proteomes from specific subcellular spaces in living cells. However, these enzymatic approaches require genetic engineering to introduce exogenous enzymes for intracellular labeling,

hindering their utilities in cells from primary samples such as native tissues and clinic samples[6], which are crucial samples that are directly linked to mechanistic dissection of biological or pathological processes. Conventional subcellular fractionation or organelle isolation methods[7] could be applied to these samples for ex situ subcellular proteomics, but require cell lysis processing (e.g., shearing force, detergent) which would fundamentally alter the native state of cells and organelles[8], and also suffer from large material input, poor spatiotemporal resolution as well as inevitable contamination. Therefore, in situ proteomic dissection of these important primary living samples remains a long-standing but challenging goal.

We envision that a chemical approach that is able to in situ label the proteome with high spatiotemporal resolution without the need of genetic manipulation, might be able to address this unmet need.

[1]Synthetic and Functional Biomolecules Center, Key Laboratory of Bioorganic Chemistry and Molecular Engineering of Ministry of Education, Beijing National Laboratory for Molecular Sciences, College of Chemistry and Molecular Engineering, Peking University, Beijing, China. [2]Peking-Tsinghua Center for Life Sciences, Peking University, Beijing, China. [3]These authors contributed equally: Ziqi Liu, Fuhu Guo. ✉e-mail: pengchen@pku.edu.cn; xinyuanfan@pku.edu.cn

Recently, a series of proximity labeling approaches[9–12] based on photocatalysts in the replacement of enzymes have been reported, but largely restricted to extracellular or ex situ applications (e.g., cell surface, isolated nuclei) as well as demonstrations on model cell lines. Alternatively, by coupling the emerging bioorthogonal photocatalytic decaging chemistry[13,14] with an activable labeling probe Quinone Methide (QM), we conceived a chemical system, termed CAT-Prox[15]. This non-genetic system leveraged the subcellular-targeting photocatalyst for living cells, which activated a chemical probe for intracellular proximity protein labeling in a photocatalytic manner. However, investigation of the more complex and clinically relevant primary living samples was still unachieved due to the efficiency limitations of the labeling warhead QM probe, which was the key component in the chemical labeling system.

Herein, we present the CAT-S system (Fig. 1), a state-of-the-art transfection-free proximity labeling technique for efficient in situ capture of mitochondrial proteome of primary cells and tissue samples. Through the development, a chemical labeling warhead, thioquinone methide (thioQM), was created and introduced as a more efficient and accurate proximity labeling probe. Combining with the thoroughly evolved photocatalytic labeling system, subcellular proteome profiling has been largely improved in terms of efficiency and accuracy, enabling the dissection of the mitochondrial proteome of diverse samples including hard-to-transfect cells or primary cells from living tissues as well as clinical blood specimens, which otherwise remain inaccessible by current methods. Furthermore, the high labeling efficiency and negligible sample perturbations of our CAT-S system allowed the identification of previously uncharacterized mitochondrial proteins as well as subcellular protein atlas of diseases within their native states, providing a landscape view as well as molecular insights of the primary and living samples from physiological or clinical settings.

## Results

### Establishment of the CAT-S system

We started by the investigation on the chemical labeling warhead, which was the key factor for assuring the labeling efficiency and sensitivity. As a reactive Michael receptor, QM has been previously reported as a reactive intermediate for the covalent labeling of protein residues[16]. However, the QM-based proteome capture[15] displayed significantly lower efficiency than the enzymatic methods, resulting in inadequate proteomic coverage and moderate specificity, likely due to their relatively lower protein-labeling reactivity than the reactive intermediate in enzymatic approaches such as phenolic radicals in APEX[5]. We turned to tune the reactivity of QM probes by adapting the chemical structures[17]. Notably, the substitution of the oxygen atom of QM with sulfur atom would yield a special type of QM species, thioQM[18], which has been an underdeveloped reactive species but might possess much higher reactivity attributing to the following properties: (i) faster formation of the thioQM intermediate upon the decaging reaction due to the less electronegativity of the sulfur atom than the oxygen atom (2.4 vs 3.5) and (ii) more stable state of the reactive zwitterionic resonance due to the larger atomic radius of sulfur (0.88 Å vs 0.48 Å), making the thioQM with $10^2$–$10^3$-fold higher reactivity than the canonical QMs in nucleophilic addition reactions[16,18], which might enhance the labeling efficiency and specificity due to faster reaction. In addition, thioQM would also act as a soft electrophile given the high polarizability of the sulfur atom, which favors reactions with soft nucleophiles (e.g., protein residues[19]) over aqueous medium, potentially increasing the labeling efficiency. Given these significant enhancements, we reasoned that thioQM might hold great promise as an intriguing warhead for labeling chemistry on proteins.

For the chemical evolution of labeling warheads, we designed and synthesized an array of activatable probes based on QM and thioQM

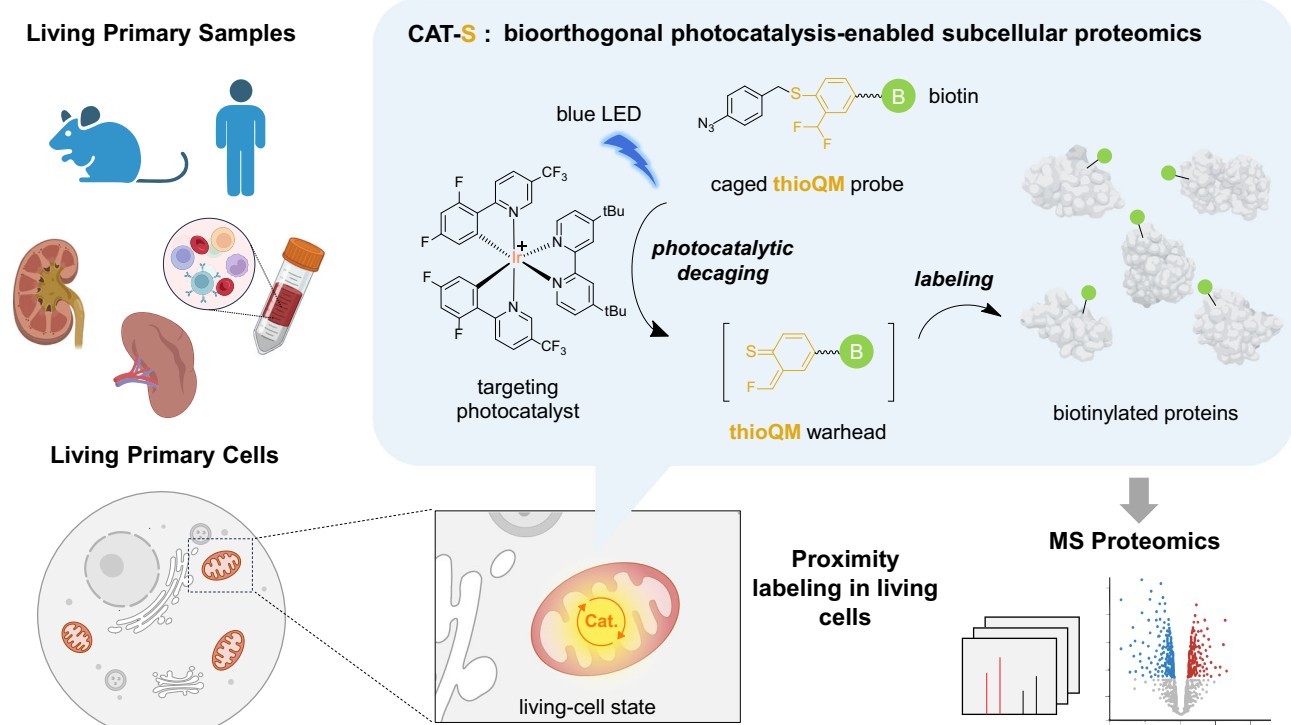

**Fig. 1 | Schematic illustration of CAT-S.** CAT-S is an in situ subcellular proteomics strategy that utilizes the cell-permeable Ir-based photocatalyst to target mitochondria in living cells from primary tissue or blood samples, which can catalyze the blue light-triggered bioorthogonal decaging reaction of the *p*-azidobenzyl (PAB) group to generate reactive thioQM probe for in situ proximity labeling of the mitochondrial proteome. No genetic engineering is required for CAT-S, rendering it widely applicable for various primary and living samples. Created with BioRender.com (agreement number JH26IJNP4R).

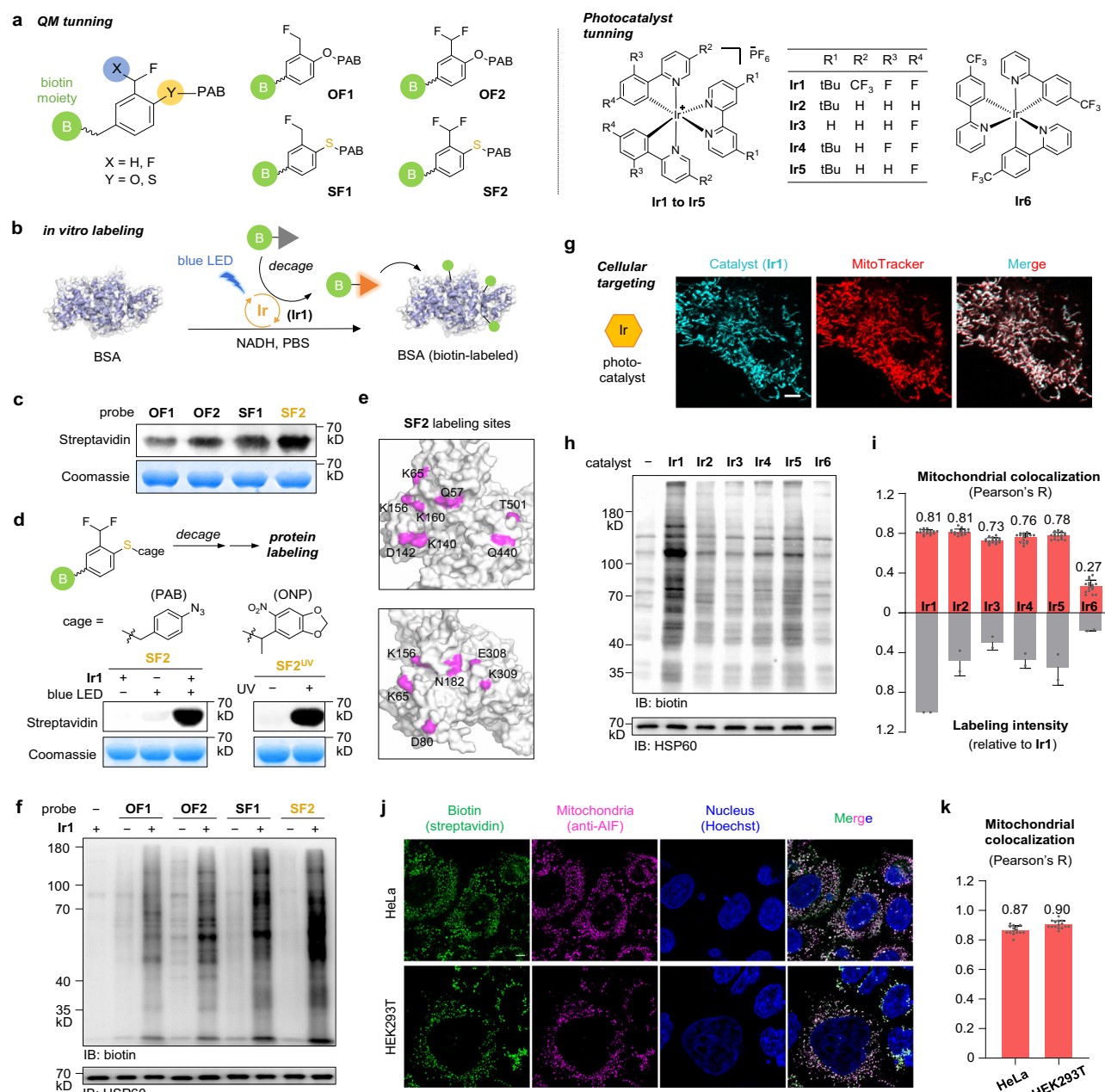

**Fig. 2 | Establishment of the CAT-S system. a** Structural tunning of biotin-tethered QM/thioQM-based activatable probes and iridium photocatalysts. **b** Schematic view of in vitro photocatalytic labeling. **c** Comparison of in vitro labeling efficiencies between different QM probes. BSA was treated with **Ir1** (5 μM), QM probes (100 μM), and NADH (500 μM) for labeling (450-nm blue LED, ~4 mW/cm², 5 min) ($n = 2$ biological replicates). **d** Decaging chemistries-controlled protein labeling by using the **SF2** warhead, as exampled by photocatalytic and UV-triggered decaying, respectively ($n = 2$ biological replicates). **e** Potential photocatalytic labeling sites for **SF2** on BSA as identified by LC–MS/MS. **f** Comparison of in cellulo labeling efficiencies between different probes. HeLa cells were treated with 100 nM **Ir1** and 100 μM probe for labeling. Mitochondrial protein HSP60 was blotted as the loading control ($n = 2$ biological replicates). **g** Confocal imaging of live HeLa cells treated with **Ir1** (0.5 μM, blue) and MitoTracker Deep Red (100 nM, red). Scale bar, 5 μm

($n = 2$ biological replicates). **h** Comparison of catalytic labeling efficiencies between different photocatalysts. HeLa cells were treated with photocatalyst (100 nM) and **SF2** probe (100 μM) for labeling ($n = 2$ biological replicates). **i** Quantification of mitochondrial colocalization (Pearson's $R$ value, $n = 15$ cells) and catalytic labeling efficiency (anti-biotin blotting intensity, related to panel (**h**)). Data are presented as mean ± SEM. **j** Immunofluorescent images of HeLa (top) and HEK293T (bottom) cells after CAT-S labeling (100 nM **Ir1** and 100 μM **SF2**). Green, labeling signal detected by streptavidin. Red, staining of mitochondrial marker AIF. Blue, staining of the nucleus by Hoechst 33342. Scale bar, 5 μm ($n = 2$ biological replicates). **k** Quantification of mitochondrial colocalization (Pearson's $R$ value) for CAT-S labeling. Data are presented as mean ± SEM ($n = 15$ cells). Source data are provided as a Source Data file.

(Fig. 2a). The phenolic oxygen or sulfur atom in these probes were protected by *para*-azidophenyl (PAB) group, which can be removed by the bioorthogonal photocatalytic decaging chemistry, releasing QM or thioQM upon light trigger to label proximal proteins in situ. The tethered biotin tag further enabled the detection or isolation of the captured proteins. The in vitro labeling efficiency was first investigated

by using bovine serum albumin (BSA) as the model protein (Fig. 2b). Using the commercially available photocatalyst **Ir1** and additive NADH, we observed significant labeling within minutes upon mild blue light irradiation for all the probes (Fig. 2c and Supplementary Figs. 1 and 2). To our delight, the thioQM probes exhibited higher labeling efficiencies than the commonly used oxygen-QM probes as expected,

with the difluoro-substituted thioQM probe (**SF2**) showing the highest labeling intensity, likely due to the further increased electrophilic reactivity by introducing electron-withdrawing fluorine atom. Given that the arylazide group might be activated through photochemistry to generate other reactive species and side labeling[20], we replaced the PAB group with a routinely used protecting group ONP[21] (removable by UV light) by synthesizing **SF2ᵘᵛ** probe without arylazide moiety for comparison (Fig. 2d). Decaging of **SF2ᵘᵛ** upon UV irradiation led to similar intensive labeling as the **SF2** upon photocatalytic decaging, suggesting that the labeling was dominantly dependent on thioQM but not the arylazide moiety. By LC–MS/MS analysis after purifying the modified peptides from the thioQM-labeled BSA[22], we identified multiple potential sites with expected thioQM modifications (Fig. 2e and Supplementary Fig. 3). Similar to the previously reported QMs[23], the **SF2**-derived thioQM warhead exhibited promiscuous reactivity to a wide range of nucleophilic protein residues (eight amino acids observed: Lys, Tyr, Asp, Asn, Glu, Gln, Trp, and Thr), with Lys being labeled most frequently likely attributing to its high reactivity coupled with surface exposure. All the detected sites were located on the protein surface, while the internally buried residues were not observed to be labeled.

We next investigated the applicability of these probes for capturing proteins in living cells (Fig. 2f). After sequential incubation with photocatalyst **Ir1** and QM probe, the living HeLa cells were irradiated by blue LED to activate the QM probe for in situ protein labeling. Indeed, all four probes exhibited significant protein labeling in the presence of photocatalyst **Ir1**, among which **SF2** probe again presented the highest labeling efficiency (Fig. 2f). We also examined the reported arylazide probe (**AzPh**)[24] as a non-QM control, with nearly no labeling observed in contrast to the intensive labeling by **SF2**, confirming the QM-dependent labeling by CAT-S inside living cells (Supplementary Fig. 4). Therefore, we selected **SF2** as the optimal labeling probe for further investigations.

Since the photocatalyst is crucial for both mitochondria specificity and catalytic activity, a systematic evaluation of the photocatalyst was conducted next, in terms of mitochondria targeting and photocatalytic activity synergistically (Fig. 2g–i). By tracking the fluorescent signals of various iridium photocatalysts in living cells, we observed significant localization of cationic compounds **Ir1** - **Ir5** in mitochondria (Fig. 2g and Supplementary Fig. 5). In contrast, the neutrally charged compound **Ir6** displayed nearly no targeting ability, indicating that the positive charge played an essential role for the mitochondrial targeting, due to the electrochemical gradient known as mitochondrial membrane potential (MMP)[25,26]. Next, we performed the photocatalytic decaging chemistry using **SF2** probe both in vitro and in living cells to evaluate the photocatalytic activity (Fig. 2h and Supplementary Fig. 6). Remarkably, photocatalyst **Ir1**, which bears a *tert*-butyl-bipyridine ligand and two difluorophenyl-(trifluoromethyl)pyridine ligands, displayed much higher labeling activity than the other catalysts, suggesting that the electron-withdrawing group on phenylpyridine ligand might enhance the catalytic performance. Taken together, with its outstanding performance on both mitochondrial targeting specificity as well as photocatalytic activity in living cells, **Ir1** was chosen as the optimal photocatalyst for the CAT-S system (Fig. 2i). Finally, by immunofluorescence detection, we visualized the **Ir1**-catalyzed labeling signals inside cells, with the biotinylation signal precisely colocalized with the mitochondria marker (Fig. 2j, k). Statistical analysis further revealed high Pearson's $R$ values (0.87 for HeLa and 0.90 for HEK293T) of colocalization, confirming the spatial specificity of our labeling system.

## Validation and evaluation of the CAT-S system

With the chemical system established, we sought to capture mitochondrial proteomes in living cells with in-depth dissection by mass spectroscopy (Fig. 3, Supplementary Fig. 7, and Supplementary Data 1), especially for those samples difficult for enzymatic approaches. As an initial verification, we applied our CAT-S system in cancer cell line HeLa by using **SF2** as the labeling probe and **Ir1** as the photocatalyst (Fig. 3a–c, top row). Living Hela cells were collected after 12-min irradiation by mild blue LED (450 nm, ~4 mW/cm²) and the biotinylated proteins were enriched by streptavidin beads (Supplementary Fig. 8). A negative control was made by omitting the catalyst. After LC–MS/MS analysis for the (+) catalyst and (−) catalyst groups with thousands of proteins detected, we applied ratiometric cutoff of +/− catalyst ratios based on the distribution of true- and false-positives to filter off background signals, in order to retain denoised high-confidence proteome (~80% specificity) (Fig. 3b, top). The final protein list for HeLa across biological triplicates revealed 391 enriched proteins, including 326 known mitochondrial proteins (83% specificity) according to MitoCarta3.0[27] and UniProt[28] database (Fig. 3d). Surprisingly, the mitochondrial proteins accounted for 95% of the detected protein MS abundance, suggesting the highly efficient capture of mitochondria proteome by our method (Fig. 3e). In contrast, previous chemical method CAT-Prox (187 mitochondrial proteins, at 71% specificity)[15] and mitochondria-localizable reactive molecules (MRMs, 259 mitochondrial proteins, at 69% specificity)[29] displayed much lower mitochondrial protein coverage as well as specificity, indicating the significant advancement of the current CAT-S system in terms of labeling efficiency and sensitivity (Fig. 3f).

Next, in order to compare the efficiency of CAT-S with the enzymatic technology, we chose the genetic operation-accessible human embryonic kidney 293T (HEK293T) cells for evaluation (Fig. 3a–c, middle row). Using the same CAT-S protocol and cutoff threshold as for HeLa cells, we obtained a protein list of 422 proteins including 334 known mitochondrial proteins (79% specificity, accounting for 96% of protein abundance) (Fig. 3d). The results showed both increased specificity and mitochondrial protein number than the previously reported enzymatic studies[30] using biotin ligases BioID (255, 70% specificity), TurboID (209, 67% specificity), or miniTurbo (269, 75% specificity), though inferior to the highly efficient mito-APEX method (464, 94% specificity)[5] probably due to the longer lifetime of QM intermediate.

Given the independence of genetic operation, the CAT-S technique is advantageously accessible to hard-to-transfect cells. Therefore, we next evaluated CAT-S with K562 cells[31] (Fig. 3a–c, bottom row), which are hematopoietic leukemia known to be difficult for genetic modification. Immunoblotting analysis on K562 cells applied with CAT-S displayed significant biotinylation signals, indicating successful photocatalytic labeling. Notably, LC–MS/MS analysis yielded a list of 386 proteins containing 336 known mitochondrial proteins (87% specificity, accounting for 92% of protein abundance) (Fig. 3d). Combined analysis of the aforementioned three cell lines indicated that the captured mitochondrial proteomes were largely overlapped (Fig. 3g and Supplementary Fig. 9), with proteins responsible for essential pathways in mitochondria (Fig. 3h and Supplementary Fig. 10). Sub-mitochondrial profiles of the captured mito-proteomes suggested the preference of CAT-S to label mitochondrial matrix proteins (Fig. 3i), in accordance with the MMP-based mitochondrial targeting ability of **Ir1**. Each of the three proteomes covered 36–37% of the entire human mitochondrial matrix proteins. The marginal capture of inter-membrane space and outer membrane proteins might be attributed to extended probe diffusion facilitated by pore structures on inner membranes (e.g., permeability transition pore[32]), or background noises. Collectively, the successful applications to K562 and other cells validated the generality of CAT-S to portray mitochondrial proteomes for various cell types, especially the hard-to-transfect cells that are inaccessible by enzymatic approaches.

Additionally, to address the concern of potential photo-cytotoxic effects by photocatalyst, we assessed the biocompatibility of our

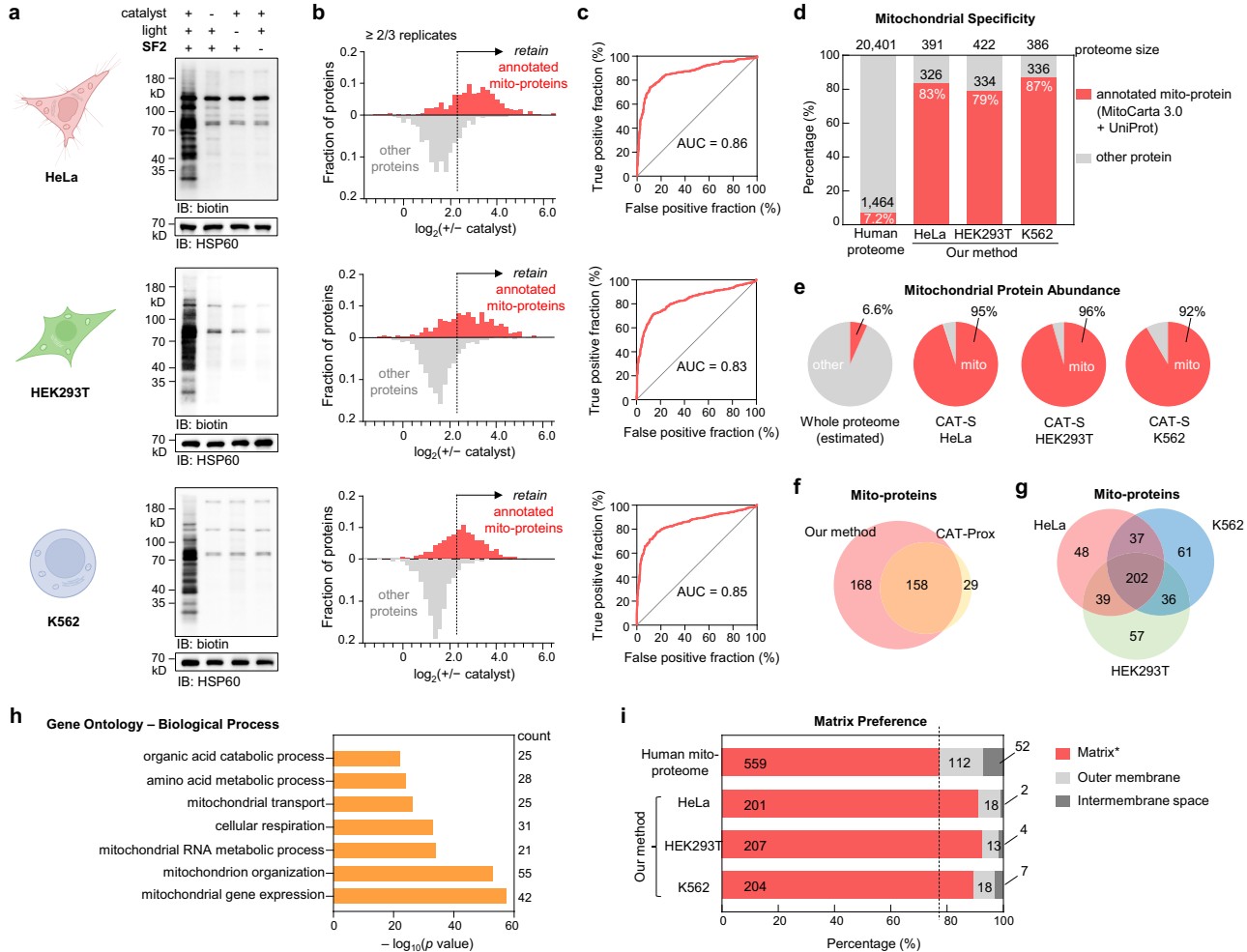

**Fig. 3 | Validation and evaluation of the CAT-S system. a–c** CAT-S-enabled in situ mitochondrial proteomics of living HeLa, HEK293T, and K562 cells (from top to bottom), treated with/without **Ir1** and **SF2** followed by blue LED irradiation. **a** Labeling signals measured by immunoblotting ($n = 3$ biological replicates). **b** Frequency distribution profiles of MS-identified proteins (≥2/3 replicates) with (red) and without (gray) mitochondrial annotation according to the combined human MitoCarta3.0 and UniProt database. A filter based on +/− catalyst ratio (set to 5.0 for the three cell lines) was further applied to subtract background noise (non-specific proteins). **c** ROC curves of CAT-S for detecting annotated mito-chondrial proteins, with AUC (area under the curve) values shown. **d** Mitochondrial specificity analysis for HeLa, HEK293T, and K562 cells with the columns showing the fraction of proteome with prior mitochondrial annotation in the database. **e** Contribution of annotated mitochondrial proteins (red) in the total protein MS

abundance. The estimated abundance of mitochondrial proteins in the whole cellular proteome is shown for comparison. **f** Comparison of mitochondrial proteomic coverage between this work and CAT-Prox. **g** Venn diagram showing the overlap between captured mitochondrial proteomes. **h** GO biological process enrichment analysis of the mutually captured mitochondrial proteome from HeLa, HEK293T, and K562. Top terms ($P < 10^{-20}$, cumulative hypergeometric test) were shown. **i** Sub-mitochondrial preference analysis of captured mitochondrial proteomes from HeLa, HEK293T, and K562 cells. Proteins were classified into "matrix*", "inter-membrane space (IMS)", and "mitochondrial outer membrane (MOM)" according to MitoCarta3.0 database. The "matrix*" category also contains the matrix-facing subunits of inner membrane protein complexes according to the literature[5]. Source data are provided as a Source Data file. Created with BioRender.com (agreement number LJ26KFYU6T).

system (Supplementary Fig. 11). After performing CAT-S to label mitochondrial proteome in living cells, we collected the cells for pro-liferation assay and MMP measurement, to evaluate cellular and mitochondrial health, respectively. Gratifyingly, all three cell lines (HeLa, HEK293T, and K562) after CAT-S labeling stayed alive and retained the same proliferation activity as the control group (Supple-mentary Fig. 11a, b), demonstrating that CAT-S is a non-toxic proximity labeling technique. Accordingly, the cells largely maintained the MMP after CAT-S labeling, suggesting that the integrity and activity of mitochondria were also preserved (Supplementary Figs. 11c and 12). Furthermore, "Two-round proteome labeling" was also conducted by applying CAT-S on the same living cell sample twice, further confirm-ing the high biocompatibility of our CAT-S system (Supplementary Fig. 11d). Given these advantages, we envisioned that CAT-S might further enable dynamic pulse-chase applications, such as the study of protein translocation between subcellular spaces[33], which require to

maintain cell growth after initial labeling followed by the secondary subcellular capture process.

## "Mito orphans" discovered by CAT-S

Besides the known mitochondrial proteins, several non-mitochondrial proteins according to the MitoCarta3.0 and UniProt database were frequently observed by our CAT-S system, which prompted us to re-investigate their locations inside cells (Fig. 4 and Supplementary Figs. 13 and 14). Through analysis for cell lines, we identified a fraction of proteins as "mito orphans" without prior mitochondrial annotation but clustered with known mitochondrial proteins (Fig. 4a, Supple-mentary Data 2, and Supplementary Software).

Applying the well-established validation strategy[5], we investigated the subcellular location of potential human "mito orphans" by expressing "orphan" proteins in HEK293T cells for imaging. To our delight, three proteins (PTPN1, SLC35A4 upstream ORF (uORF), and

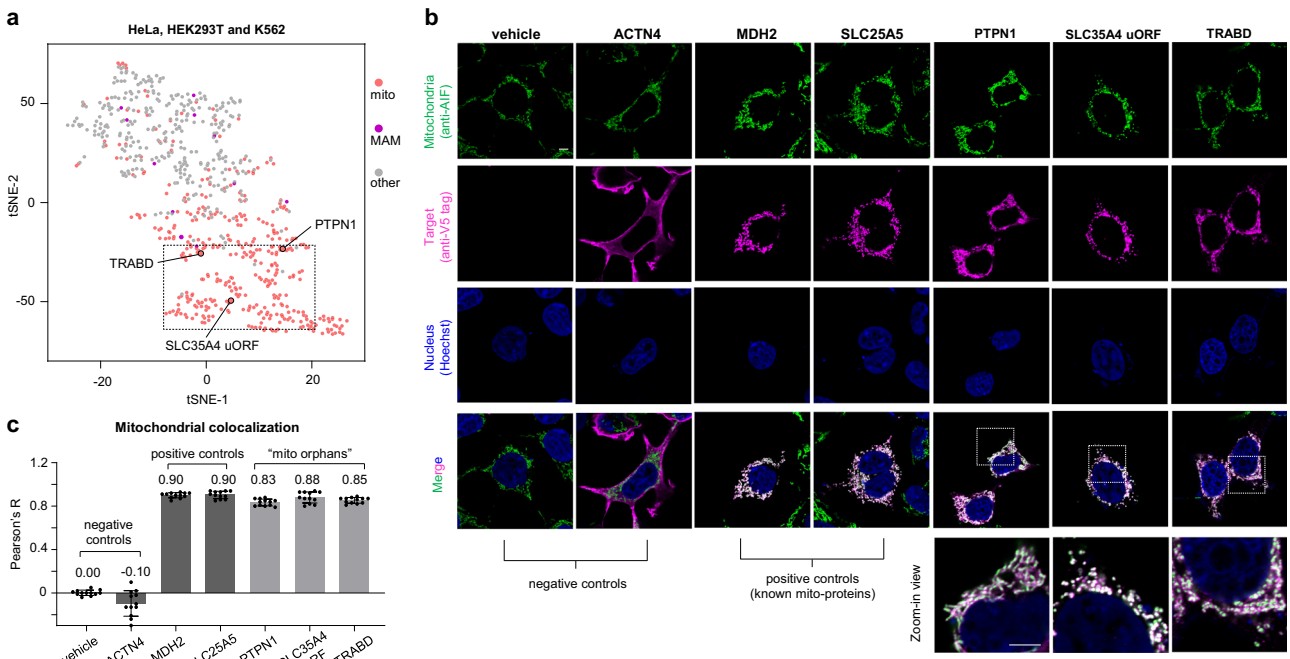

**Fig. 4 | CAT-S-guided discovery of "mito orphans". a** Visualization of CAT-S datasets including $\log_2(+/-$ catalyst) values of individual experiments in two dimensions by t-SNE analysis, for combined (HeLa, HEK293T, and K562) datasets. Proteins are colored based on their prior annotations in the database. Red, mitochondrial localization. Purple, MAM (mitochondria-associated membrane) or ER-mitochondria contact site localization. Proteins without such annotation but clustered with mitochondrial proteins are potential "mito orphans". The validated "mito orphans" are denoted. **b** Imaging of overexpressed orphan proteins in HEK293T cells. Green, staining of mitochondrial marker AIF. Red, staining of overexpressed target protein by C-terminal V5 tag. Blue, staining of the nucleus by Hoechst 33342. Scale bar, 5 μm. **c** Quantification of mitochondrial colocalization (Pearson's *R* value) for "mito orphans" and controls. Data are represented as mean ± SD (*n* = 12 cells). Source data are provided as a Source Data file.

TRABD) were identified as mitochondrial proteins to the best of our knowledge, with predominant colocalization with mitochondria (Fig. 4b, c). Given its function as a phosphatase, PTPN1 might play a role in the signal transduction related to mitochondria. The transmembrane proteins SLC35A4, uORF, and TRABD, which were highly uncharacterized proteins without clear knowledge of functions in the database, might be potentially intriguing targets for further investigation. Additionally, we also identified TMEM160 in our analysis, and initially categorized this protein as a "mito orphan" (Supplementary Fig. 13). A recent study[34] published during our research period has concurrently confirmed human TMEM160 to be mitochondrial, further validating the efficacy of CAT-S-guided mitochondrial discovery. These findings indicated the in situ profiling by CAT-S could also shed light on the hidden members of the mitochondrial proteome.

## CAT-S enabled mitochondrial proteome profiling for primary cells from tissues

Capturing subcellular proteomes in living tissues is highly desirable yet challenging. Applying the enzymatic proximity labeling approaches to tissues normally requires the construction of the transgenic animal model, which could be time-consuming and often inapplicable in many cases (e.g., clinical human tissues). Given the transfection-free advantage and the biocompatibility of CAT-S, we sought to capture mitochondrial proteomes directly from primary cells of living tissues (Fig. 5, Supplementary Fig. 15, and Supplementary Data 3). Since human metabolism and immunity critically rely on mitochondria, kidney- and splenocyte-related diseases have been often found in association with mitochondrial dysfunctions[35,36], making dissection of mitochondrial protein networks crucial for mechanistic understanding. We chose to apply our CAT-S system in mitochondrial proteome profiling on the kidney and spleen, the highly metabolic organ and the essential immune organ, respectively. As demonstrated, the tissues were

excised from healthy B6 mice and disassociated into cell suspensions with red blood cells removed, and then subjected to CAT-S procedure for mitochondrial proteomic labeling and LC–MS/MS analysis. Quality control measurement by western blotting displayed significant biotinylation signals after photocatalytic labeling (Fig. 5b). Due to the distinct properties between tissue types, we performed a cutoff analysis based on labeling extent for kidney and spleen, respectively, to obtain denoised proteomic lists of considerable confidence (>60% specificity). Across biological replicates, we finally revealed a list of 430 proteins (261 known mitochondrial proteins, 61% specificity) for mice kidney and 357 proteins (228 known mitochondrial proteins, 64% specificity) for mice spleen (Fig. 5c–f). The proteomes covered 20–21% of the total known mitochondrial matrix proteins, spanning the essential mitochondrial functions. Analysis with other cutoff criteria yielded largely identical proteomes as well as functional profiles (Supplementary Fig. 16). The moderately lower coverage and specificity of captured proteomes from tissues than from immortalized cell lines might be attributed to the high heterogeneity and fragility of these primary samples. Further analysis revealed that mitochondrial proteins contributed 75–80% of the protein abundance, indicating the effective mitochondrial proteomic capture from distinct primary living tissues (Fig. 5e). Although we could not exclude the potential stress brought by photocatalytic labeling on tissue cells, the overall profiles of their mitochondrial proteomes may remain unaffected since the CAT-S labeling period was short (10-min scale), in contrast to the long protein turnover time (generally from hours to days)[37,38] in tissue cells.

Seeking biological insights on mitochondria at the tissue level, we performed Gene Ontology (GO) analysis for our captured proteins (Fig. 5g, h and Supplementary Fig. 16), especially those only detected in post-cutoff proteomes of kidney (Fig. 5g) or spleen (Fig. 5h). For "kidney-only" dataset, aerobic respiration was ranked as the most enriched process, suggesting the preference of kidney mitochondria

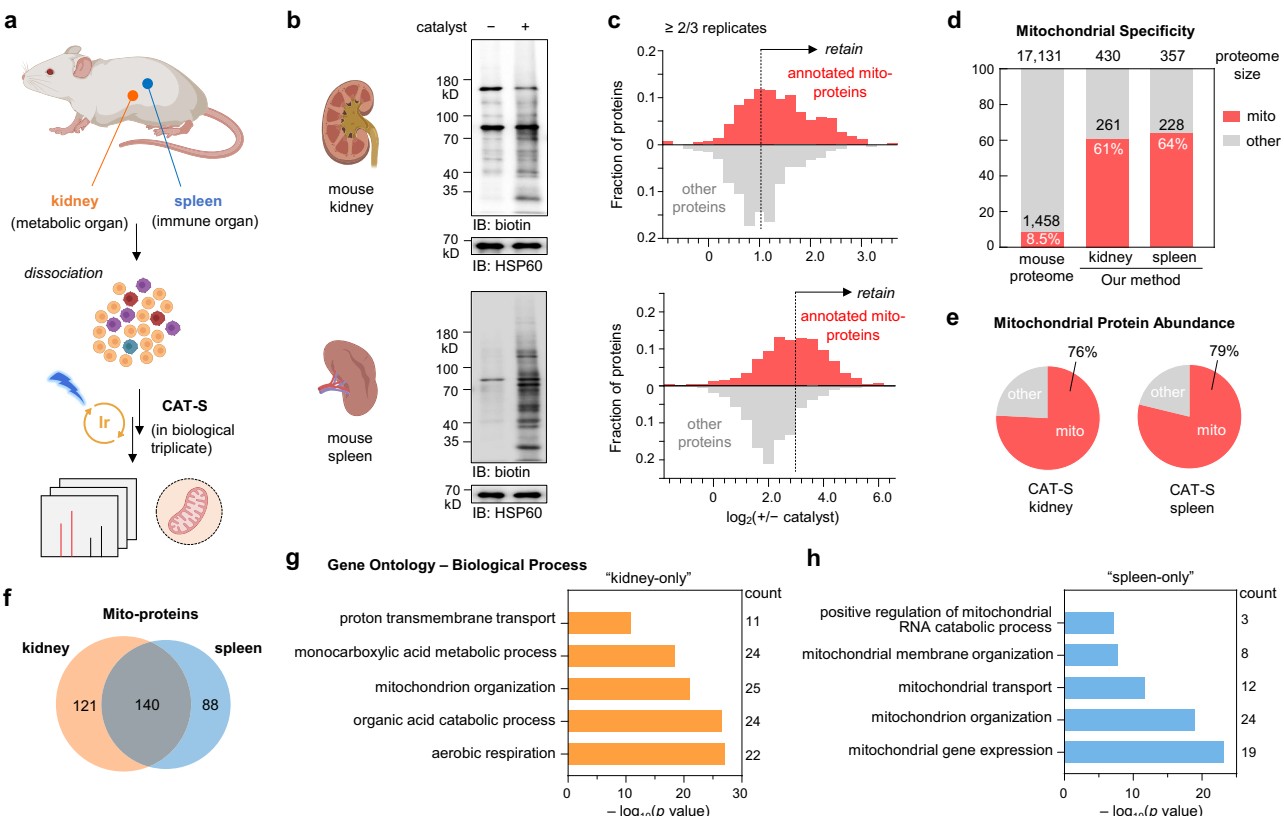

**Fig. 5 | Mitochondrial proteome profiling for primary living tissues. a** Schematic view of CAT-S workflow for mouse tissues. **b–c** CAT-S-enabled mitochondrial proteome profiling for mouse kidney and spleen, treated with **Ir1** and **SF2** followed by blue LED irradiation (*n* = 3 biological replicates). **b** Labeling signals of the experiment and control groups measured by immunoblotting analysis. **c** Frequency distribution profiles of MS-identified proteins (≥2/3 replicates) with (red) and without (gray) mitochondrial annotation according to the combined mouse Mito-Carta3.0 and UniProt database. A filter based on +/− catalyst ratio (set to 2.0 for the kidney and 8.0 for the spleen, see also Methods section) was further applied to subtract noise (non-specific proteins). **d** Mitochondrial specificity analysis for mouse kidney and spleen with the column showing the fraction of proteome with prior mitochondrial annotation. **e** Contribution of annotated mitochondrial proteins (red) in the total protein MS abundance. **f** Venn diagram showing the overlap between the post-cutoff proteomes. **g, h** Gene Ontology analysis (biological processes) for the "kidney-only" (**f**) and "spleen-only" (**g**) proteomes. The top five enriched biological processes are shown, with *p* values calculated by cumulative hypergeometric test. Source data are provided as a Source Data file. Created with BioRender.com (agreement number FE26IJO5WS).

to energy metabolism (Fig. 5g), which is consistent with kidney's role as the second highest oxygen consumer in body at rest[35]. Moreover, enzymes related to various small molecule metabolic pathways were enriched in kidney mitochondria, underpinning the kidney's role as a highly metabolic organ. In contrast, spleen samples were significantly enriched with proteins related with mito-gene expression, mitochondrial organization, and mitochondrial transport pathways (Fig. 5h), suggesting the higher activity of mitochondrial biogenesis and morphological regulation in spleen. Given that the spleen accommodates a huge quantity of immune cells (predominantly B cells and T cells), our observation indicated the elevated mitochondrial biogenesis and dynamics within this organ that might be required by immune cell development and migration[39]. Cross-analysis with the reported in-depth tissue proteomic data[40], showed that our captured "kidney-only" and "spleen-only" proteomes were typically more significant in the reported dataset for the corresponding tissue (Supplementary Fig. 17), further supporting the proteomic characteristics observed by CAT-S.

## Mitochondrial proteome atlas for the diabetic disease enabled by CAT-S
The global diabetes pandemic, especially obesity-induced type-2 diabetes (T2D), continues to grow as a severe threat to human health[41]. As a metabolic disorder, diabetes usually leads to various health complications including cardiovascular and kidney diseases, which are closely linked to mitochondrial dysfunctions[35,42]. However, the molecular mechanisms underlying the diabetes-associated diseases remain largely unclear. To this end, we sought to apply CAT-S for profiling the native state of the mitochondrial proteomes from living kidneys in diabetic settings (Fig. 6, Supplementary Fig. 18, and Supplementary Data 4).

We utilized the well-documented *db/db* mice as the obesity-induced T2D model[43], and *m/m* mice as the nondiabetic group for comparison (Fig. 6a, b). The kidneys from 10-week-old obese-diabetic and healthy mice were excised, and subjected to CAT-S procedure as described above. Protein labeling was confirmed by immunoblotting, with labeling intensities at similar levels on both samples (Fig. 6c). To quantitatively compare the mitochondrial proteomes from disease and healthy kidneys as well as filter out background signals, we used triplex dimethyl labeling for MS-based relative quantification. We first performed a cutoff analysis based on +/− catalyst ratios to remove obvious non-biotinylated proteins (filter 1), and then retain the proteins detected both in diabetic and non-diabetic datasets (filter 2). Finally, we intersected the protein lists from biological triplicate to generate a high-confidence protein list (Fig. 6b), consisting of 400 proteins including 281 known mitochondrial proteins (70% specificity) (Fig. 6d, e). The data was further visualized as a volcano plot (Fig. 6f), showing over one hundred up- and down-regulated mitochondrial proteins from diabetic mouse kidneys.

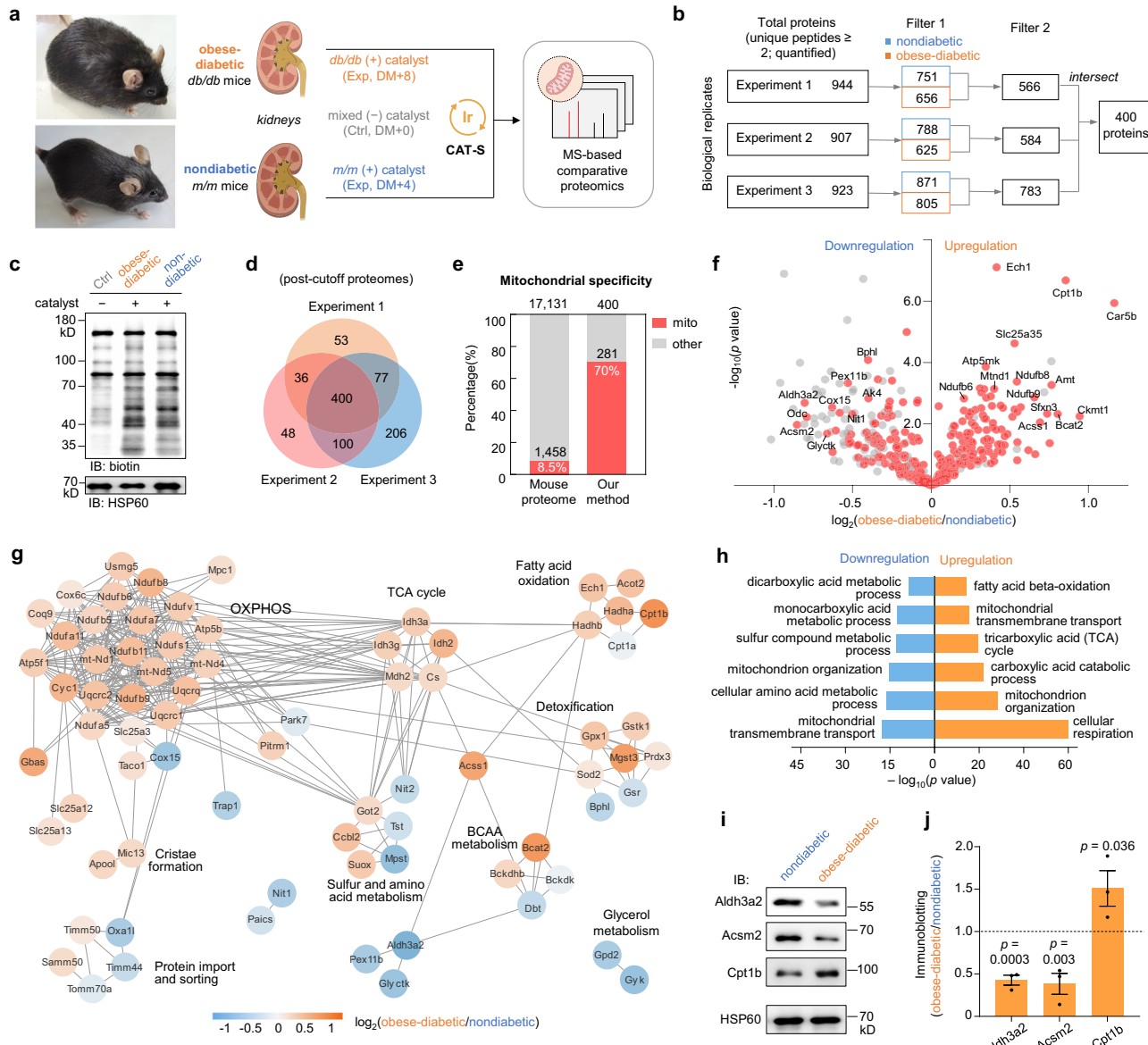

**Fig. 6 | Comparative mitochondrial proteomics for diabetic and healthy mice.** **a** Schematic view of CAT-S-enabled comparative mitochondrial proteomics for diabetic and nondiabetic mouse kidneys. Samples were treated with **Ir1** and **SF2** followed by blue LED irradiation. **b** Filtering of MS data to produce a protein list for comparative proteomics. First column from left, proteins detected with ≥2 unique peptides and quantified ratios by MS. Second column, retained proteins after cutoff based on diabetic/Ctrl or nondiabetic/Ctrl ratios (set to 1.2) to remove non-biotinylated proteins while maximizing the retention of labeled proteome (Filter 1). The diabetic and nondiabetic protein lists were intersected to produce lists for relative quantification (Filter 2). Lists from three independent experiments were intersected to generate the final proteome. **c** Labeling signals measured by immunoblotting analysis. **d** Venn diagram showing the overlap between the comparative proteomes of three experiments. **e** Mitochondrial specificity analysis with bars showing the fraction of proteome with prior mitochondrial annotation (red) in the database. **f** Volcano plot showing the regulation of proteins in diabetic and nondiabetic mouse kidney ($n = 3$ biologically replicates). Red dots, proteins with prior mitochondrial annotation. $p$ Values were calculated by unpaired two-tailed $t$ test. **g** Cluster map showing regulations of functionally associated molecular complexes according to STRING analysis[61] and Markov clustering. Gray lines denote protein–protein interactions with high confidence (interaction score > 0.7). **h** Top six enriched biological processes for up- and down-regulated mitochondrial proteins in obese-diabetic mouse kidney by Gene Ontology analysis, with $p$ values calculated by cumulative hypergeometric test. **i**–**j** Validation for three lipid-metabolism mitochondrial proteins by immunoblotting. **i** Blot images. **j** Intensity quantification. Data are represented as mean ± SEM ($n = 3$ biological replicates) with $p$ values shown (unpaired one-tailed $t$ test). Source data are provided as a Source Data file. Created with BioRender.com (agreement number HV26IJOCW8).

The identified differentially regulated proteins spanned diverse biological functions and pathways. We conducted multiple bioinformatic investigations including mapping, clustering, and enrichment analysis to elucidate biological processes and functional networks from our data (Fig. 6g, h and Supplementary Fig. 18). For upregulated mitochondrial proteins in obese-diabetic kidney (Fig. 6h, right), the cellular respiration process was highly enriched, in harmony with the elevated oxygen consumption to sustain the high energy demand in

early diabetic kidney disease[44]. According to the clustering analysis based on protein–protein interaction (PPI) (Fig. 6g) and mitochondrial pathways (Supplementary Fig. 18c), we clearly observed the elevation of oxidative phosphorylation (OXPHOS) complexes in obese-diabetic group, along with the upregulations of TCA-cycle, glucose oxidation, fatty acid oxidation (FAO), and amino acid catabolism, concurrently underlying the hypermetabolic phenotypes in the early diabetic kidney[35].

Interestingly, we also detected apparent alterations in lipid metabolism, mitochondrial transmembrane transport, detoxification, and other pathways, with complicated regulations under obese-diabetic conditions. Given the importance of lipid metabolism in diabetes[45], we further chose three lipid-metabolism enzymes (Aldh3a2, Acsm2, and Cpt1b) with significant changes according to proteomic profile for further validation (Fig. 6i, j and Supplementary Fig. 19). Using both whole-cell lysate and mitochondrial lysate as samples, we discovered significant upregulation (Cpt1b) and downregulations (Aldh3a2 and Acsm2) for obese diabetic kidney, consistent with the CAT-S proteomic data. The upregulated protein, Cpt1b, is a member of the carnitine palmitoyl transferase family known for synthesizing acylcarnitine for FAO and is also reported to be inversely related to insulin sensitivity[46]. Among the downregulated proteins, Acsm2 is known for synthesizing medium-chain fatty acyl-CoA for metabolic utilization and found with decreased expression in abnormal kidney[47]. Given these previous hints, we suspected that the reshaping of the lipid metabolic network might play essential and multi-faceted roles during diabetic nephropathy. Particularly, the aberrant decrease of Aldh3a2 and Acsm2, which are enzymes responsible for converting fatty aldehydes and medium-chain/xenobiotic fatty acids, respectively, would lead to the accumulation of such lipids and consequent cellular toxicities (e.g., metabolism disruption, biomolecule damage by aldehydes, and signaling alterations)[48–50]. Thus, we concluded that the obese-diabetic condition could suppress the detoxification machinery in lipid metabolism, therefore driving the progression of kidney disorders and diseases. Additionally, we also observed that Cpt1a, an isoform of Cpt1b that catalyzes the same reaction, stayed nearly unchanged (diabetic/non-diabetic ratio = 0.94) while Cpt1b and other enzymes related to FAO were generally upregulated. Given that Cpt1a predominates in lipogenic tissues (e.g., liver) while Cpt1b predominates in tissues with high FAO capacity (e.g., heart and skeletal muscle)[51], we speculated that the increased Cpt1b/Cpt1a ratio under obese-diabetic condition might also shift the kidney to be more like a fatty acid consumer, in accordance with the reported hypermetabolic phenotypes[35]. Noteworthy, these molecular features have not been revealed before in proteomic investigations of diabetic kidney[52], which may provide insights for our understanding of the pathological processes, further demonstrating the advantages of our CAT-S technique for portraying in situ mitochondrial landscapes in primary living tissue samples.

## Mitochondrial proteomics for primary T cells from human PBMC

Applying the CAT-S system for clinically relevant samples, such as cells from human peripheral blood that is critical for practical diagnosis, would be another exciting direction for deciphering human biology and diseases. Among the cell types in peripheral blood mononuclear cells (PBMC), T cells are critical in adaptive immunity as well as diverse diseases including cancers, infections, and autoimmunity, rendering the characterization of human blood T cells in high demand for diagnosis and mechanistic study. Moreover, mitochondria have been found to play fundamental and multi-faceted roles underlying diverse T cell processes[39], particularly on activation[53,54] and exhaustion[55], with the molecular characteristics yet resolved.

Given that primary T cells and other immune cells are difficult for transfection, we reasoned that CAT-S could be applied to T cells from clinical blood samples for in situ mitochondrial proteomics (Fig. 7 and Supplementary Data 5)[6]. To obtain pure and living T cells, we firstly separated PBMCs from the whole blood by density-gradient centrifugation, and then performed magnetic negative cell sorting to isolate primary T cells (Fig. 7a). Immunoblotting assay showed that 20 nM **Ir1** generated intensive protein labeling (Fig. 7b), suggesting the high sensitivity of CAT-S for T cell mitochondrial proteome labeling. With the limited volume of the clinical blood samples, we used ~$3 \times 10^7$ primary T cells (<0.5 mg total cellular protein due to the small cellular size) in each labeling and control group for MS analysis. We performed CAT-S experiments using samples from three healthy donors. The mitochondria-specific labeling was confirmed by the distribution profile (Fig. 7c). After cutoff analysis to reduce non-specific background, the protein lists of three donors were obtained, with 219 known mitochondrial proteins detected from more than two donors (62% specificity), which accounted for 75% of the detected protein abundance (Fig. 7e, f).

GO analysis further elucidated the major pathways (Fig. 7g) and functions (Fig. 7h) involved by our identified mitochondrial proteins in primary T cells from PBMC, exhibiting a profile similar to our dataset on mice splenocytes (Supplementary Fig. 15d). Cellular respiration and mitochondrion organization ranked as the most enriched biological processes, in accord with their indispensable roles for cell homeostasis. Interestingly, in contrast to the results on fast-proliferating cell lines mentioned above (Supplementary Fig. 10), the process of mitochondrial gene expression was detected with relatively low enrichment for primary T cells, possibly due to the quiescent state of T cells under normal conditions. Although our method herein was demonstrated by using healthy human samples, further application of CAT-S for comparative mitochondrial profiling between disease and healthy conditions, or between different stages of T cell activation, would unveil more critical roles of mitochondria for immunobiology[56].

## Discussion

In summary, we have developed a transfection-free chemical platform CAT-S for proximity labeling in primary living cells, which leveraged the advantages of the bioorthogonal photocatalytic decaging methods and thioQM-mediated labeling chemistry. By coupling with in-depth MS-based proteomics, CAT-S was applied to a variety of living cell samples, including multiple cell lines, dissociated mouse tissues as well as primary T cells from human blood, providing mitochondria-centric views of proteome landscape as well as relative quantification between disease-related conditions.

Through the development and application of CAT-S, intriguing insights into both chemistry and biology studies have been unveiled. During the chemical exploration, we developed an activatable warhead, thioQM, which showed excellent proximity labeling efficiency that enabled the otherwise difficult protein capturing in living primary samples. The reactive thioQM could be generated on demand by various decaging strategies (e.g., photocatalytic or UV decaging), showing potential for diverse applications involving covalent labeling. By profiling mitochondrial proteomes of distinct cell types, we meanwhile revealed a set of mitochondrial located or associated proteins, including three proteins (phosphatase PTPN1, transmembrane proteins SLC35A4 uORF, and TRABD) that were predominantly located in mitochondria. Moreover, by quantitatively comparing mitochondrial proteomes between obese-diabetic and nondiabetic kidneys, we observed major alterations of pathways involving lipid metabolism, with metabolic enzymes such as Cpt1b, Acsm2, and Aldh3a2 differentially regulated, suggesting that obese-diabetic condition suppresses the detoxifying lipid metabolic machinery to progress the diabetic nephropathy.

The CAT-S system exhibited a set of advantages as a state-of-the-art in situ mitochondrial proteomics technique: (i) CAT-S enabled in situ mitochondrial proteome capture in diverse living cell samples including hard-to-transfect cells, primary cells from tissues and blood samples that are otherwise inaccessible by genetic engineering and enzymatic approaches; (ii) by introducing the thioQM warhead for efficient protein labeling, CAT-S significantly enhanced the capture efficiency and specificity over previous chemical methods; (iii) CAT-S showed high biocompatibility with no obvious cellular toxicity under the experiment conditions, in contrast to the methods with harsh treatment; and (iv) CAT-S is flexibly controlled by external light to turn

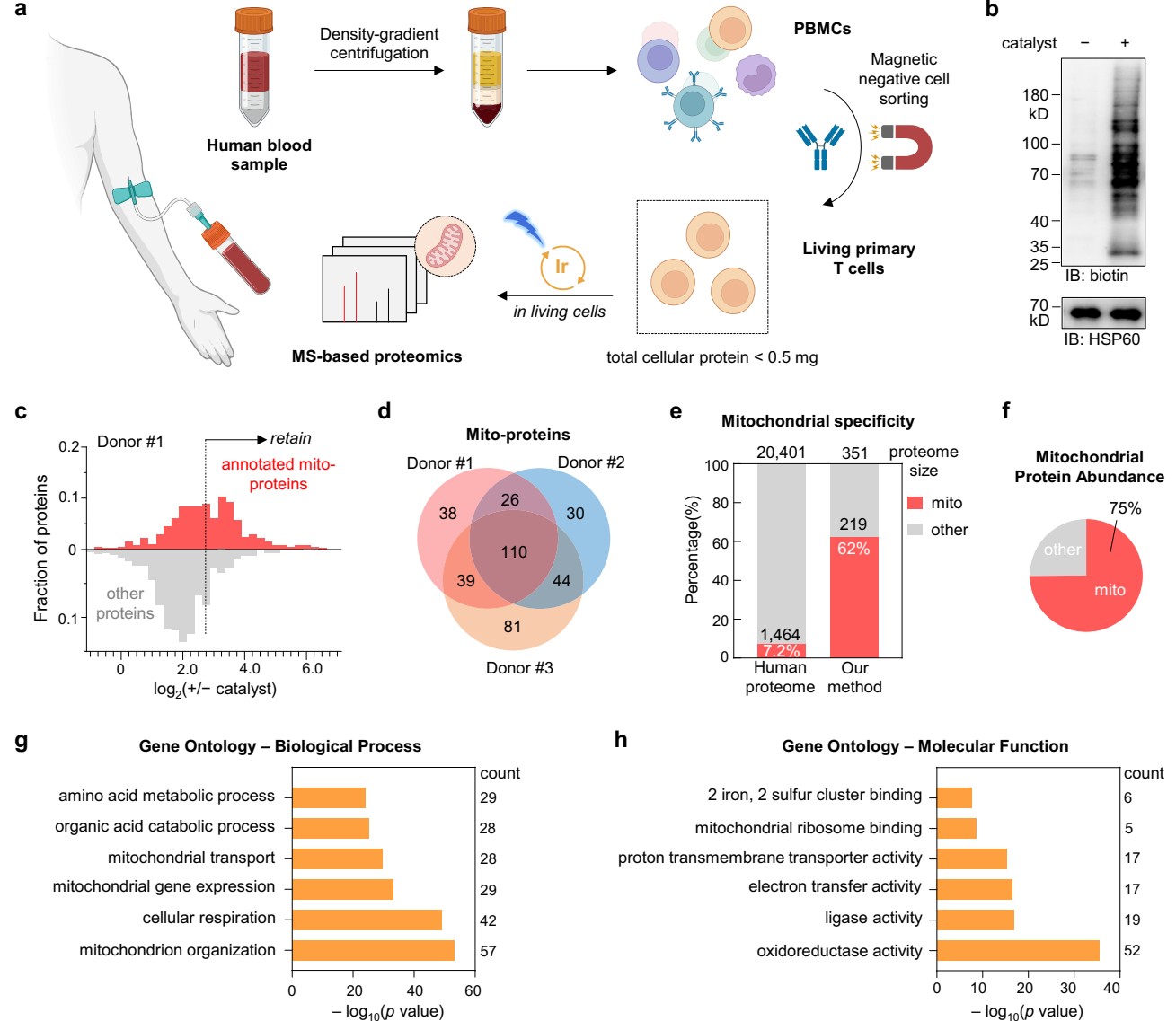

**Fig. 7 | Mitochondrial proteomics for primary T cells from human peripheral blood. a** Schematic view of CAT-S mitochondrial proteomics for living primary T cells from human peripheral blood. PBMCs, peripheral blood mono-nuclear cells. Samples were treated with **Ir1** and **SF2** followed by blue LED irradiation (*n* = 3 independent samples). **b** Labeling signals measured by immunoblotting analysis. **c** CAT-S-enabled mitochondrial proteome profiling of primary T cells, as examined by Donor#1. Left, frequency distribution profiles of MS-identified proteins with (red) and without (gray) mitochondrial annotation according to the combined human MitoCarta3.0 and UniProt databases. A filter based on +/− catalyst ratio (set to 7.0) for further noise subtraction (non-specific proteins) is illustrated. **d** Venn diagram showing the overlap between captured mitochondrial proteomes of the three donors. **e** Mitochondrial specificity analysis for the captured proteins (detected from two or more donors), with columns showing the fraction of proteome with prior mitochondrial annotation (red). **f** Contribution of annotated mitochondrial proteins (red) in the total protein MS abundance. **g**, **h** Gene Ontology analysis for the captured mitochondrial proteins (detected from two or more donors). The top six enriched biological processes (**g**) and molecular functions (**h**) are shown, with *p* values calculated by cumulative hypergeometric test. Source data are provided as a Source Data file. Created with BioRender.com (agreement number ME26KGQEMI).

on/off the photocatalytic labeling system, with more precise control of labeling time-window than the substrate-controlled enzymatic methods, enabling high temporal resolution for potential dynamic investigations. Indeed, limitations also exist for the CAT-S method in comparison to enzymatic approaches. The coverage and mitochondrial specificity were observed lower than the phenolic radical-based method APEX, probably due to the longer lifespan of QM intermediate compared to phenolic radical. The potential hydrophobic interactions (e.g., with lipid droplets in adipose samples) as well as leakage of small-molecule iridium catalyst may also compromise the spatial precision, while the genetically encoded enzymes, especially the recently emerged super-resolution proximity labeling methods[57–59], could enable highly precise spatial localization and proteomic capture in

cells. Despite these limitations, the non-genetic CAT-S strategy, with its advantages and potential for further engineering, stands as a valuable addition to the proximity labeling toolbox. This is especially pertinent for research involving primary living samples.

Future applications and extensions of the CAT-S strategy can be foreseen in multiple aspects. For methodology development, CAT-S can be further coupled to the continuously evolving LC–MS/MS techniques and bioinformatics, thus enabling higher sensitivity and deeper dissection of mitochondria as well as other subcellular locations. Further improvement of sample processing and analytical methods, such as optimizing enrichment workflow or applying modification-based analysis from super-resolution techniques[57–59], would also enhance the resolution of CAT-S and reduce the ambiguity

between target proteins and noises. Indeed, given that small molecule photocatalysts could be chemically engineered to target various sub-cellular spaces[60], further adaption of the CAT-S system to profile other subcellular proteomes is currently under investigation in our laboratory. For further applications, CAT-S is advantageously accessible to living primary samples, and investigations on their in situ mitochondrial proteomes of different stages (e.g., T cell activation and exhaustion), disease conditions, or therapeutic effects (e.g., before/after immune checkpoint therapy) by CAT-S can also be foreseen. A complete chemical toolbox for the unprecedented mapping of subcellular-resolved proteomics of intact primary samples with excellent spatiotemporal resolution is expected, which will greatly facilitate our understanding of the native molecular mechanism for biological and pathological processes from real primary samples.

## Methods

### Ethical statement

All animal studies were approved by the Institutional Animal Care and Use Committee of Peking University (CCME-ChenP-3). All protocols and experiments using human blood samples from Oricells (Shanghai OriBiotech Co., Ltd) were approved by the ethics committee of Shanghai Zhaxin Hospital (No. LP202006).

### Materials

**General considerations.** All expression media, buffers, and antibiotics were prepared using purified $H_2O$ (Mili-Q Reference) and autoclaved or filter-sterilized, as appropriate. Photocatalysts were commercially purchased from J&K Scientific® or Energy Chemical®. Protein and nucleic acid concentrations were determined by NanoDrop 2000 spectrophotometer (Thermo). Images of protein gels including nucleotide agarose gel, coomassie SDS-PAGE gel, and western blotting membranes were taken on ChemiDoc XRS+ (Bio-Rad) with Image Lab Touch software (v2.3.0, BioRad). UV-vis spectrometry for cell samples was measured on a Synergy H4 microplate reader (Bio-Tek) with Gen5 software (v3.11.19, BioTek). Confocal microscopy images were obtained on an LSM 700 laser scanning confocal microscope (Zeiss) with ZEN software (v2.1, ZEISS). Flow cytometry was performed on LSRFortessa cell analyzer (BD biosciences) supplemented with BD FACSDiva (v7.0, BD biosciences) software. Figures with cartoons were created with BioRender.com.

**Cell culture.** HeLa (catalog#1101HUM-PUMC000011), HEK293T (catalog#1101HUM-PUMC000091), and K562 (catalog#1101HUM-PUMC000039) cells were purchased from Cell Resource Center, Peking Union Medical College, China. HeLa and HEK293T cells were grown in DMEM (Dulbecco's modified Eagle's medium, Gibco) supplemented with 10% fetal bovine serum (FBS, Gibco) at 37 °C in 5% $CO_2$. K562 cells were grown in RPMI 1640 supplemented with 10% fetal bovine serum (FBS, Gibco) at 37 °C in 5% $CO_2$.

**Mouse model.** C57BL/6 J (B6) mice (female, 6–8-week, Cat# 219) were purchased from Beijing Vital River Laboratory Animal Technology (Beijing), China. *db/db* obese-diabetic mice (male, 7–8-weeks) and *m/m* nondiabetic mice (male, 7–8-week) were purchased from Cavens Laboratory Animal Technology (Changzhou), China. The three *db/db* mice (41.1, 40.5, and 46.6 g) and three *m/m* mice (26.5, 27.1, and 26.7 g) were weighed before execution. All the mice were housed at 18–24 °C with 40–70% humidity and a 14-h light/10-h dark cycle.

**Human sample.** Human blood samples from three healthy donors (male; aged 30, 21, and 27, respectively) were purchased from Oricells (Shanghai OriBiotech Co., Ltd). The human PBMCs were isolated from the blood sample using an isolation kit (Solarbio, Cat#P8610) according to the manufacturer's protocol.

**Molecular cloning.** Molecular cloning operations were all carried out according to the manufacturer's protocols. Primers were ordered from GENEWIZ Biotech (Suzhou). Human cDNA was synthesized by using Hifair® 1st strand cDNA synthesis superMix (YEASON, 11141ES) and human RNA extracted from HEK293T cells by TRIzol reagent. PCR was performed using Phanta Max Super-Fidelity DNA polymerase (Vazyme, P505-d1) in 50 µL reactions. In vitro, homologous recombination was carried out using Exnase MultiS (Vazyme, C113-02) in 20 µL reactions (with ~120 ng linearized vector and ~60 ng insert). Digestion of template was performed using DpnI (Thermo, ER1701) in Tango buffer.

### In vitro labeling assays

**Model protein labeling.** Photocatalyst (5 µM) was combined with PAB-caged QM/thioQM probes (100 µM), NADH (500 µM), and bovine serum albumin (BSA, 1 mg/mL) in PBS (pH 7.4) to afford reaction mixtures with total solution volume of 100 µL. The samples were placed on a 96-well plate (100 µL/well) and irradiated by mild blue LED (450 nm, 4 mW/cm²) (unless otherwise noted) at room temperature for 5 min (unless otherwise noted). Eighty microliter of each sample was taken and combined with 20 µL 5× SDS-PAGE non-reducing loading buffer (Beyotime). Immunoblotting was performed with 4–15% gradient SDS-PAGE and 0.2 µm PVDF membrane. Streptavidin-HRP (Solarbio, SE068, 1:1,000 dilution) was used to detect the biotinylated bands. For validation of photocatalytic labeling by **SF2**, groups without **Ir1** or without light irradiation were used as controls.

For UV-triggered labeling, **SF2**[UV] (100 µM) was combined with BSA (1 mg/mL) in PBS (pH 7.4) to afford reaction mixtures with a total solution volume of 100 µL. The samples were placed on a 96-well plate (100 µL/well), irradiated by UV light (CL-1000 UVP Ultraviolet Cross-linker) for 10 min at room temperature, and analyzed by immunoblotting as described above.

### Identification of labeling site by LC–MS/MS

**Protein labeling and isolation.** The reaction was performed by combining **Ir1** (10 µM), **SF2**-alkyne (100 µM), NADH (2 mM), and BSA (2 mg/mL) in PBS buffer (pH 7.4) with a total solution volume of 2 mL. The reaction system was placed in a 6-well plate and irradiated by mild blue LED (4 mW/cm²) for 15 min at room temperature. The BSA protein was then precipitated by mixing the solution with cold methanol and chloroform (solution:methanol:chloroform = 4:4:1, v/v/v), followed by >30 min of freezing at −80 °C. The sample was centrifuged (10,000 g, 20 min, 4 °C) and washed by cold methanol twice to yield a white pellet of protein.

**Click reaction.** The BSA pellet was redissolved in PBS containing 1.2% SDS (200 µL solution per sample) under sonication and heated to 90 °C for 5 min, followed by dilution with PBS to reach the point of 0.2% SDS. The solution was then mixed with 2 mM Cu-BTTAA (1:2, prepared by mixing CuSO$_4$ and BTTAA in $H_2O$), 3 mM sodium ascorbate, and 2 mM DADPS (cleavable biotin-azide, BBBD-19, Confluore Biological Technology). After the Click reaction at 37 °C for 1 h with shaking, the protein in solution was purified by methanol-chloroform precipitation protocol as described above.

**Streptavidin enrichment.** The protein pellet precipitated from the Click reaction was redissolved in 1.2% SDS-PBS and then diluted with PBS to reach the point of 0.2% SDS. High-capacity streptavidin resin beads (Thermo, 20357) were equilibrated by washing three times with 1 mL PBS. Bead slurry (100 µL) was added and the mixture was incubated for 3 h at room temperature with gentle rotation. After the incubation, the beads were pelleted by centrifugation (1,400 g, 3 min), washed by 0.2% SDS-PBS (5 mL, 10 min incubation) once, and further washed by PBS (5 mL) five times.

**Thiol blocking and trypsin digestion.** The protein-bound streptavidin beads were pelleted by centrifugation (1,400 g, 3 min) and the supernatant was removed, followed by the addition of 500 μL 6 M urea in PBS and 25 μL 200 mM dithiothreitol (DTT). The beads were resuspended by vortexing and incubated at 37 °C for 30 min in a Thermo-Mixer (Eppendorf). Twenty-five microliter of 400 mM iodoacetamide (IAA) was then added, and the resulting mixture was incubated again at 37 °C for 30 min in the dark. After incubation, beads were washed with 0.1 M trimethylammonium bicarbonate (TEAB) buffer (1 mL, four times) and pelleted. Two-molar urea–PBS buffer was added to the beads, followed by the addition of 2 μg trypsin. The beads were incubated at 37 °C for 16 h in a ThermoMixer to digest the proteins.

**Isolation of modified peptides by acidic cleavage.** After on-bead trypsinization, the supernatant was removed, and the beads were washed by PBS (1 mL, three times) and H₂O (1 mL, three times). The beads were then incubated with 2% formic acid (200 μL) in a ThermoMixer for 1 h at room temperature, and the supernatant was collected. The procedure was repeated once. The beads were further washed with 50% MeCN/H₂O (with 1% formic acid) twice, with the supernatant collected. All the collected supernatant fractions were combined and evaporated to dryness in a vacuum concentrator.

**LC–MS/MS data acquisition and analysis.** Peptides were dissolved in 0.1% formic acid and analyzed on Orbitrap Fusion Lumos LC–MS (Thermo) coupled to a Dionex Ultimate 3000 RPLC nano system (Thermo), with Thermo Xcalibur (v4.1.50) software. Samples were loaded onto a loading column (100 μm × 2 cm) and a C18 separating capillary column (75 μm × 15 cm) packed in-house with Luna 3-μm C18(2) bulk packing material (Phenomenex, USA). The HPLC solvent A was 0.1% formic acid in H₂O, and the solvent B was 0.1% formic acid in 80% MeCN. The samples were run using a 120-min gradient method (0 min 2% B; 8 min 2% B; 9 min 10% B; 123 min 44% B; 128 min 99% B; 138 min 99% B; and 139 min 2% B) with a flow rate of 300 nL/min. The Fusion Lumos was operated in data-dependent mode with a cycle time of 3 s. The MS1 was performed with a full-scan $m/z$ range from 350.00 to 1,600.00 and a mass resolution of 60,000. The maximum injection time was set to 50 ms with an AGC target of 4e5. The MS/MS fragmentation was performed using a quadruple isolation window of 1.6 $m/z$ and the HCD collision mode (30% energy), with the orbitrap resolution set to 15,000, maximum injection time set to 30 ms, and AGC target set to 5e4. Collected data were analyzed using pFind (version 3.1.5) software. Mass spectra were searched against the BSA (UniProt ID P02769) protein sequence, with mass tolerance set to 20 ppm and max missing cleavage (by trypsin) number set to 3. The minimum peptide length was set to 7. Carbamidomethylation at Cys (+57.0215 Da) was set as a fixed modification. Oxidation at Met (+15.9949 Da) and putative SF2-QM-IAA (+509.2108 Da) modification at all nucleophilic residues (Lys, Arg, Ser, Thr, Tyr, Asp, Glu, Asn, Gln, Trp, Met, His, and Cys) were set as variable modifications. The putative arylazide-labeling (+575.2378 Da) modification was also searched. Identified peptides were manually reviewed. The Open Search mode was turned off.

### Imaging of photocatalyst subcellular location

Approximately 10,000 HeLa cells were seeded in a LabTek-II 8-well glass chamber with 200 μL DMEM (+10% FBS) and incubated at 37 °C with 5% CO₂ for 24–48 h. The medium was removed and the cells were rinsed with PBS. **Ir1** (0.5 μM) was combined with MitoTrackerTM Deep Red (0.1 μM, Thermo) in 100 μL DMEM and the cells were incubated with the mixture for 30 min at 37 °C. After washing by PBS three times, the cells were immediately analyzed o an LSM 700 laser scanning confocal microscope (Zeiss). The fluorescent images were captured in the Hoechst channel (λex = 405 nm) for photocatalyst luminescence and the Cy5 channel (λex = 639 nm) for MitoTracker using a 63× oil

lens. Processing of images were carried out using ZEN 3.2 blue edition (ZEISS). Pearson's $R$ values for colocalization were calculated for individual cells using the Coloc2 tool embedded in Fiji-Image J (v1.53k).

### In cellulo CAT-S for miscellaneous assays
**Photocatalytic labeling in living cells**
**Adherent cells.** Cells (e.g., HeLa, HEK293T) were cultured to >90% confluence in a 12-well plate (approximately 10⁶ cells/well). After removal of medium and washing with PBS (1 mL/well), cells were incubated with photocatalyst (**Ir1** or other, 100 nM in 1 mL DMEM) for 30 min at 37 °C with 5% CO₂, followed by washing with PBS once. Fresh DMEM (1 mL/well) was added, and the cells were incubated for 15 min at 37 °C with 5% CO₂. After removal of the medium, a labeling probe (**SF2** or other, 100 μM in 0.5 mL DMEM) was added and the cells were incubated for 30 min at 37 °C with 5% CO₂, followed by mild blue LED irradiation (~4 mW/cm², 12 min) at room temperature and incubation in the dark for 10 min at 37 °C. After further washing with PBS twice, the cells were ready for subsequent assays.

**Suspension cells.** The protocol was adjusted with additional transfer and centrifugation steps. Cells (e.g., K562) were cultured to near saturation in the T25 or T75 flask, with approximately 2 × 10⁶ cells used for each sample. After removal of the medium by centrifugation (300–500 g, 3 min, 4 °C) in a tube, the cells were washed with PBS (1 mL/tube), followed by resuspension in RPMI-1640 containing photocatalyst (**Ir1** or other, 50 nM in 1 mL medium. Note: 20–50 nM catalyst is sufficient for suspension cells, due to their higher uptake efficiency than adherent cells). The culture was transferred to a 12-well plate and incubated at 37 °C with 5% CO₂ for 30 min (all the incubation steps herein were performed in a well plate or culture dish). Then the cells were transferred back to the tube for centrifugation to remove media, washed with PBS, and resuspended in RPMI-1640 (1 mL/sample), followed by incubation for 15 min at 37 °C with 5% CO₂. After removal of the medium, the cells were resuspended in RPMI-1640 with a labeling probe (**SF2** or other, 100 μM in 0.5 mL medium), incubated for 30 min at 37 °C with 5% CO₂, followed by mild blue LED irradiation (~4 mW/cm², 12 min) at room temperature and incubation in the dark for 10 min at 37 °C. After further washing with PBS twice, the cells were ready for the subsequent assays.

**Immunoblotting assay.** For sample preparation, adherent cells after photocatalytic labeling were directly scrapped from plate using RIPA strong lysis buffer (120 μL/well) (CWBIO, CW2333S) supplemented with 1× protease inhibitor cocktail (Bimake, B14001) and transferred to 1.5-mL tube, while suspension cells were pelleted by centrifugation (300–500 g, 3 min, 4 °C) and lysed in the same buffer. The samples were incubated on ice for 10 min, followed by the addition of methanol and chloroform (lysate:methanol:chloroform = 4:4:1, v/v/v) to precipitate proteins for purification. After cooling the mixture for >30 min at −80 °C, the proteins were pelleted by centrifugation (10,000 g, 20 min, 4 °C), carefully washed with cold methanol (200 μL) twice, and redissolved in 1% SDS-PBS buffer (80 μL). Five times SDS-PAGE loading buffer (20 μL) (Beyotime) was added to the sample, followed by heating at 95 °C for 15 min. Immunoblotting was performed with Tris-glycine SDS-PAGE and 0.2 μm PVDF membrane. Mouse anti-biotin mAb (Santa Cruz, sc-101339, 1:1,000 dilution) and rabbit anti-HSP60 mAb (Abcam, ab45134, 1:2,000 dilution) were used as primary antibodies, while HRP-linked anti-mouse IgG (Cell Signaling Technology, 7076S) and HRP-linked anti-rabbit IgG (Cell Signaling Technology, 7074S) were used as secondary antibodies (1:5,000 dilution).

**Immunofluorescence assay.** For photocatalytic labeling, cells (HeLa, HEK293T) were seeded on a LabTek-II 8-well glass chamber (pre-coated with Matrigel, Corning 356234), cultured to 50–80% confluency, and subjected to the CAT-S photocatalytic labeling procedure

as described above. The incubation and washing steps were performed with a liquid volume of 200 µL/well. Upon completion of labeling, cells were washed with PBS (200 µL/well, twice), and fixed with 4% formaldehyde for 15 min at room temperature. After one PBS wash, cells were permeabilized by 0.2%Triton-X100 in PBS for 20 min, washed by PBST solution (0.1% Tween-20 in PBS pH 7.4) three times, and blocked by 3% BSA (PBST solution) for 1 h at room temperature. With three PBST washes between each incubation step, the cells were incubated with primary antibody rabbit anti-AIF mAb (Abcam, ab32516, 1:500 dilution) at 4 °C overnight, followed by incubation fluorescent antibodies goat anti-Rabbit-AlexaFluor546 (Invitrogen, A-11010, 1:500 dilution) combined with streptavidin-AlexaFluor488 (Invitrogen, S11223, 1:500 dilution) and Hoechst 33342 (2 µg/mL) at room temperature for 1.5 h. After three PBST washes, the samples were subjected to microscopy analysis on an LSM 700 laser scanning confocal microscope (Zeiss). The fluorescent images were captured in Hoechst channel ($\lambda$ex = 405 nm), AF488 channel ($\lambda$ex = 488 nm), and AF546 channel ($\lambda$ex = 555 nm). Processing of images were carried out using ZEN 3.2 blue edition (ZEISS).

**Cell proliferation assay.** The cells after photocatalytic labeling were harvested in the 1.5-mL tube (for adherent cells, Trypsin-EDTA digestion was performed for disassociation). The cells were pelleted by centrifugation (300–500 g, 3 min, 4 °C), and resuspended in PBS (1 mL). 20 µL cell suspension was added to 100 µL culture media (DMEM or RPMI-1640) in a well in 96-well plate, performed in triplicate. The cells were further cultured at 37 °C with 5% $CO_2$ for 48 h before analysis. MTS assay was carried out using CellTiter 96 Aqueous One Solution kit (Promega, G3581) according to the manufacturer's protocol. In brief, 10 µL of One Solution Reagent was added to each sample in a 96-well plate, and the plate was incubated at 37 °C with 5% $CO_2$ for 1 h. The absorbance at 490 nm of each well was recorded on a Synergy H4 microplate reader (Bio-Tek). Controls were made by omitting photocatalysts (non-phototoxic group) during photocatalytic labeling. The proliferation data were normalized to the non-phototoxic control (100%) and medium (0%). GraphPad Prism 8.0 software was used to create the graphs and perform unpaired two-tailed Student's $t$ test.

**MMP assay.** The cells after photocatalytic labeling were harvested as described above. Samples omitting photocatalysts were used as healthy control, while samples with a five-fold concentration of **Ir1** were used as phototoxic control. After resuspension in PBS (1 mL), 200 µL cell suspension was added to a well in a V-bottom 96-well plate. After pelleting by centrifugation (400 g, 3 min, 4 °C), the cells were resuspended in 200 µL culture medium containing 2 µg/mL JC-1 MMP probe (Bioss, D-9113) and incubated for 30 min at 37 °C with 5% $CO_2$. Then the cells were washed with culture medium (200 µL) twice by centrifugation, resuspended in culture medium (200 µL), and immediately analyzed on BD LSRFortessa cell analyzer. JC-1 monomer was detected using the FITC channel, while JC-1 aggregate (forming in the presence of MMP) was detected using the PE channel. Data was further analyzed using FlowJo v10 software (FlowJo, Tree Star).

## CAT-S mitochondrial proteomics for general cells
**Photocatalytic labeling in living cells.** This labeling protocol for proteomics was the scale-up version of the procedure described in "In cellulo CAT-S for miscellaneous assays", and applied generally for living cells.

**Adherent cells.** Cells (e.g., HeLa, HEK293T) were cultured to >90% confluence in a 10-cm dish (approximately $10^7$ cells per sample). After removal of medium and wash with PBS (5 mL per dish), the cells were incubated with **Ir1** photocatalyst (100 nM in 10 mL DMEM, for HeLa and HEK293T) for 30 min at 37 °C with 5% $CO_2$, followed by wash with

PBS. Fresh DMEM (10 mL per dish) was added, and the cells were incubated for 15 min at 37 °C with 5% $CO_2$. After removal of the culturing medium, an **SF2** probe (100 µM in 5 mL DMEM) was added and the cells were incubated for 30 min at 37 °C with 5% $CO_2$, followed by mild blue LED irradiation (~4 mW/cm², 12 min) at room temperature and incubation in the dark for 10–15 min at 37 °C. After two washes with PBS (5 mL per dish), the cells were harvested using trypsin-EDTA (Gibco), resuspended in fresh medium (10 mL), and transferred to a 15-mL centrifuge tube placed on ice. The cells were washed by PBS, pelleted by centrifugation (400 g, 3 min, 4 °C) and placed on ice. The cell pellets could be either directly lysed or stored at −80 °C. An additional sample without a photocatalyst was used as a negative control.

**Suspension cells.** Cells (e.g., K562) were cultured to near saturation in a T75 flask, with 10–15 mL saturated culture used for each group (approximately $1-3 \times 10^7$ cells). After removal of the medium by centrifugation (300–500 g, 3 min, 4 °C) in a 15-mL tube, the cells were washed with PBS (10 mL per tube), followed by resuspension in RPMI-1640 containing **Ir1** photocatalyst (50 nM in 10 mL medium. Note: 20–50 nM catalyst is sufficient for suspension cells, due to their higher uptake efficiency than adherent cells). The culture was transferred to a 10-cm dish and incubated at 37 °C with 5% $CO_2$ for 30 min (all the incubation steps herein were performed in the culture dish). Then the cells were transferred back to the tube for centrifugation to remove media, washed with PBS, and resuspended in RPMI-1640 (10 mL per sample), followed by incubation for 10–15 min at 37 °C with 5% $CO_2$ in a 10-cm dish. After removal of the medium by centrifugation, the cells were resuspended in RPMI-1640 with **SF2** probe (100 µM in 5–10 mL medium), incubated for 30 min at 37 °C with 5% $CO_2$ in a 10-cm dish, followed by mild blue LED irradiation (~4 mW/cm², 12 min) at room temperature and incubation in the dark for 10 min at 37 °C. The cells were washed by PBS, pelleted by centrifugation, and placed on ice. The cell pellets could be either directly lysed or stored at −80 °C. An additional sample without a photocatalyst was used as a negative control.

**Cell lysis and protein enrichment.** For each labeling sample, 1 mL RIPA strong lysis buffer (pH 7.4, with 1% Triton X-100, 50 mM Tris, 150 mM NaCl, 1% sodium deoxycholate, etc.) (CWBIO, CW2333S) supplemented with 1× protease inhibitor cocktail (Bimake, B14001) was added to lyse the cell. The sample was sonicated to generate a clear lysate using Vibra Cell VCX Processor (SONICS). Then the protein was precipitated by the addition of methanol/chloroform (lysate:methanol:chloroform = 4:4:1) followed by >30 min of cooling at −80 °C. The proteins were pelleted by centrifugation (10,000 g, 20 min, 4 °C), and washed three times with ice-cold methanol (1 mL). The protein pellet was redissolved in 2% SDS-PBS (200 µL) under sonication and heated to 90 °C for 5 min. The sample was then diluted by PBS to the point of 0.2% SDS and transferred to a 15-mL centrifuge tube (Corning). A small fraction of the solution (20–100 µL) was taken from the sample, diluted to 100 µL, and used for BCA analysis against BSA standards, as well as immunoblotting quality control.

High-capacity streptavidin resin beads (Thermo, 20357) were equilibrated by washing three times with PBS (1 mL). Bead slurry was added according to the amount of the original cell lysate (normally 40 µL for 1.5–3 mg total protein). The mixture was incubated for 3 h at room temperature with gentle rotation. After the incubation, the beads were pelleted by centrifugation (1,400 g, 3 min), resuspended in PBS with 0.2% SDS, and incubated at room temperature for 10 min for washing. The beads were then washed with PBS (5 mL) for 6 times by resuspension and centrifugation.

**Quality control.** As a quality control measure, the remaining whole-cell lysate fraction (~80 µL) after BCA analysis was boiled at 95 °C for 20 min with 20 µL of 5× SDS-PAGE loading buffer added, to prepare

"whole-cell" samples. Meanwhile, a fraction (~10%) of the post-enrichment beads was taken, resuspended in 80 μL PBS buffer, and boiled with 2 mM biotin and 20 μL of 5× SDS-PAGE loading buffer added, to prepare "pull-down" samples.

Immunoblotting was performed with 4–15% gradient Tris-glycine SDS-PAGE gel and 0.20 μm PVDF membrane. Mouse anti-biotin mAb (Santa Cruz, sc-101339, 1:1,000 dilution) and rabbit anti-HSP60 mAb (Abcam, ab45134, 1:2,000 dilution) were used as primary antibodies, while HRP-linked anti-mouse IgG (Cell Signaling Technology, 7076S) and HRP-linked anti-rabbit IgG (Cell Signaling Technology, 7074S) were used as secondary antibodies (1:5,000 dilution). Ruby staining of pull-down samples was performed using 4–15% gradient Tris-glycine SDS-PAGE gel, and SYPRO™ Ruby protein gel stain kit (Invitrogen, S12001) according to the manufacturer's instruction.

**Protein digestion and dimethyl labeling.** The protein-bound strep-tavidin beads were transferred to a 1.5-mL clear tube (Axygen, MCT-150-C) and pelleted, followed by the addition of 500 μL 6 M urea in PBS and 25 μL 200 mM DTT. The beads were resuspended by vortexing, and incubated at 37 °C for 30 min in a ThermoMixer (Eppendorf). Twenty-five microliter of 400 mM IAA was then added for thiol blocking, and the resulting mixture was incubated again at 37 °C for 30 min in the dark. After incubation, the beads were washed with 0.1 M TEAB buffer (1 mL, four times) and pelleted. After the addition of 0.1 M TEAB buffer (100 μL) and trypsin (1 μg), the beads were incubated at 37 °C for 16 h in a ThermoMixer to digest the proteins.

To each 100 μL of the digested peptides in 0.1 M TEAB buffer were added 4 μL of 0.6 M $NaBH_3CN$ and 4 μL of 4% (v/v) stable isotopic formaldehyde (normally $CH_2O$ for control (−) catalyst group and $CD_2O$ for experimental (+) catalyst group). The resulting mixture was vor-texed immediately and incubated at room temperature for 30 min. After incubation, 16 μL of 1% (v/v) ammonia was added to quench the reaction, followed by the addition of 8 μL formic acid. After vortexing, heavy ($CD_2O$)- and light ($CH_2O$)-labeled samples were combined at a ratio of 1:1 (v/v) to generate duplex dimethyl-labeled samples, which were further desalted on C18 tips (Thermo, 87784) according to the manufacturer's protocol and evaporated to dryness in a vacuum concentrator.

**LC–MS/MS acquisition.** Each biological replicate was analyzed as a single-shot sample. Peptides were dissolved in 0.1% formic acid and analyzed on Orbitrap Fusion Lumos LC–MS (Thermo) coupled to a Dionex Ultimate 3000 RPLC nanosystem (Thermo), with Thermo Xcalibur (v4.1.50) software. Samples were loaded onto a loading column (100 μm × 2 cm) and a C18 separating capillary column (75 μm × 15 cm) packed in-house with Luna 3-μm C18(2) bulk packing material (Phenomenex, USA). The HPLC solvent A was 0.1% formic acid in $H_2O$, and the solvent B was 0.1% formic acid in 80% MeCN. The samples were run using a 120-min gradient method (0 min 2% B; 8 min 2% B; 9 min 10% B; 123 min 44% B; 128 min 99% B; 138 min 99% B; and 139 min 2% B) with a flow rate of 300 nL/min. The Fusion Lumos was operated in data-dependent mode with a cycle time of 3 s. The MS1 was performed with a full-scan $m/z$ range from 350.00 to 1,600.00 and a mass resolution of 60,000. The maximum injection time was set to 50 ms with an AGC target of 4e5. The MS/MS fragmentation was per-formed using a quadruple isolation window of 1.6 $m/z$ and the HCD collision mode (30% energy), with the orbitrap resolution set to 15,000, maximum injection time set to 30 ms, and AGC target set to 5e4.

**Data analysis.** All MS data were interpreted using MaxQuant v1.6.10 software. The quantification of heavy/light ratios was calcu-lated with a mass tolerance of 20 ppm. For protein ID identification, MS/MS spectra were searched against a human UniProt database containing 20,378 proteins. The minimum peptide length was set to 7.

Half-tryptic termini and up to two missing trypsin cleavages are allowed. Carbamidomethylation at cysteine (+57.0215 Da) and isotopic modifications (+28.0313 and +32.0564 Da for light and heavy labeling, respectively) at lysine/N-terminal were set as fixed modifications. Oxidation at methionine (+15.9949 Da), and acetylation of N-terminal (+42.0106 Da), were set as variable modifications. Each of the three biologically independent replicates was analyzed by MaxQuant sepa-rately. Contaminants and proteins identified as reverse hits were removed. Proteins detected with unique peptides <2 were also removed. Protein lists from replicates were combined, and only pro-teins detected over twice across biological triplicates were retained. Cutoff analysis based on the averaged heavy/light (experimental/control group, in which the control group was (−) catalyst) intensity ratio was further performed to filter off background noise (i.e., pro-teins with a ratio lower than the threshold were removed) while max-imizing the retention of true positives. Proteins detected only in the (+) catalyst group were also retained. Taking mitochondrial specificity as the paramount criterion, a cutoff ratio was set to 5.0 for HeLa, HEK293T, and K562 experiments, based on the distribution profiles of proteins with and without mitochondrial annotation according to MitoCarta3.0 or UniProt (2022/10/13) database, yielding high-confidence protein lists with specificity level around 80%. GO enrich-ment analysis was performed by Metascape (https://metascape.org). Additional cutoff analysis with other criteria was also performed.

### Discovery of mitochondrial proteins

**Identification of "mito orphan" candidates.** The HEK293T datasets (including $log_2$(+/− catalyst)) values from biological replicates for each protein) and combined datasets (including $log_2$(+/− catalyst) values from HeLa, HEK293T, and K562 experiments for each protein) were $z$ scored, and visualized by t-SNE analysis (T-distributed Stochastic Neighbor Embedding) in two-dimensional space, computed by t-SNE command from scikit-learn 1.1.3 Python package with perplexity set to 20 (Supplementary Software), available on GitHub at https://github.com/hefei8alex/CAT-S_scripts. According to the 2D plot, proteins "surrounded" by known mitochondrial proteins also without MAM (ER-mito contact site) localization evidence were picked as "mito orphan" candidates for further validation (listed in Supplementary Table 3).

**Expression of orphan protein.** HEK293T cells were seeded on a LabTek-II 8-well glass chamber (pre-coated with Matrigel, Corning 356234), and cultured to 20–60% confluency. Two hundred nanogram plasmid of C terminus V5-tagged orphan protein on a pcDNA3.1 vector was dissolved in 10 μL Opti-MEM and combined with 10 μL Opti-MEM containing 280 ng PEI. After incubation for 15 min, the mixture was combined with 180 μL DMEM (with 1% FBS), and incubated with the cells for 4 h for transfection. The media were then replaced by DMEM (with 10% FBS) and the cells were further cultured for 36 h before analysis.

**Immunofluorescent imaging.** After removal of media, the cells were washed by PBS, and fixed with 4% formaldehyde for 15 min at room temperature. After one PBS wash, cells were permeabilized by 0.2% Triton-X100 in PBS for 20 min, washed by PBST solution (0.1% Tween-20 in PBS pH 7.4) three times, and blocked by 3% BSA (PBST solution) for 1 h at room temperature. With three PBST washes between each incubation step, the cells were incubated with primary antibody rabbit anti-AIF mAb (Abcam, ab32516, 1:500 dilution) combined with mouse anti-V5 mAb (Biodragon, B1005, 1:1000 dilution) at 4 °C overnight, followed by incubation fluorescent antibodies goat anti-Rabbit-AlexaFluor488 (Invitrogen, A-11008, 1:500 dilution) combined with goat anti-Mouse-AlexaFluor555 (Invitrogen, A-21422, 1:500 dilution) and Hoechst 33342 (2 μg/mL) at room temperature for 1.5 h. After three PBST washes, the samples were subjected to microscopy analysis on an LSM 700 laser scanning confocal microscope (Zeiss). The

fluorescent images were captured in the Hoechst channel ($\lambda_{ex}$ = 405 nm), AF488 channel ($\lambda_{ex}$ = 488 nm), and AF555 channel ($\lambda_{ex}$ = 555 nm). Processing of images were carried out using ZEN 3.2 blue edition (ZEISS). Pearson's $R$ values for colocalization were calculated for individual cells using the Coloc2 tool embedded in Fiji-Image J (v1.53k).

## Mitochondrial proteomic profiling for living tissues

**Processing of living tissue samples.** Living tissues (kidney or spleen) were excised immediately after the mice (C57BL/6 J) were sacrificed. Each tissue/organ was transferred to a dish, washed by HBSS (or PBS), and cut into small pieces with media (RPMI-1640 with 10% FBS) added to prevent drying. The tissue pieces with media were then transferred to a 70-μm cell drainer fitted on a 50-mL tube, gently ground using a 2-mL syringe plunger, and pressed through the cell drainer, while media was continuously added during the process. The collected cells were pelleted by centrifugation (400 g, 3 min), resuspended in 0.3 mL HBSS buffer, and gently mixed with 1 mL RBC (red blood cell) lysis buffer (TIANGEN, RT122). The mixture was incubated at room temperature for 3 min to lyse red blood cells followed by the addition of 3 mL HBSS. The cells were then pelleted from the mixture by centrifugation, washed by HBSS (15 mL) once, and resuspended in media (10 mL) for subsequent applications.

## CAT-S mitochondrial proteomics

**Photocatalytic labeling.** Tissues (kidney or spleen) disassociated as described above were employed for CAT-S mitochondrial proteomics. One mouse was normally used for two labeling samples (i.e., experimental and control group). Photocatalytic labeling was performed according to the general protocol for suspension cells (described in "CAT-S mitochondrial proteomics for general cells"), with the following optimizations: (1) all the incubation steps and irradiation step were performed in a 6-cm dish with 5 mL media due to the smaller sample size; (2) **Ir1** concentration was optimized to 100 nM for kidney samples and 50 nM for spleen samples (due to the higher uptake efficiency of splenocytes), respectively.

**MS sample preparation.** Cell lysis, protein enrichment, enzymatic digestion, and isotopic labeling processes were all performed according to the general protocol described in "CAT-S mitochondrial proteomics for general cells".

**LC–MS/MS acquisition.** Samples were run under the same condition (120-min gradient method on Orbitrap Fusion Lumos LC–MS) described in "CAT-S mitochondrial proteomics for general cells".

**Data analysis.** The obtained MS data were interpreted using Max-Quant v1.6.10 software as described in "CAT-S mitochondrial proteomics for general cells", searched against a mouse UniProt database containing 17,107 proteins. Contaminants and proteins identified as reverse hits were removed. Proteins detected with unique peptides <2 were also removed. Protein lists from replicates were combined, and only proteins detected over twice across biological triplicates were retained. Cutoff analysis based on the averaged heavy/light intensity ratio (reflecting spatial accessibility to photocatalyst) was further performed to filter off background noise (proteins with a ratio lower than threshold were removed). Proteins detected only in (+) catalyst group were also retained. Taking mitochondrial specificity as the paramount criterion, the cutoff ratio was optimized to 2.0 for the kidney and 8.0 for the spleen in order to obtain high mitochondrial specificity (>60%) protein lists, based on the distribution profiles of proteins with and without mitochondrial annotation according to MitoCarta 3.0 or UniProt (2022/10/13) database. GO enrichment analysis was performed by Metasacpe (https://metascape.org). Additional

cutoff analysis with other common criteria was also performed (Supplementary Fig. 16).

For comparison and validation, the proteins detected only in post-cutoff kidney and spleen proteomes (i.e., "kidney-only" and "spleen-only") were cross-analyzed with the published datasets[40] (https://www.ebi.ac.uk/pride/archive/projects/PXD030062) of high-coverage mitochondrial proteomics for mouse tissues by Mann group, which contained six samples for each tissue. Proteins were mapped to their datasets by UniProt ID. The data were normalized in the same way as reported[2], in which the intensity value of each protein was $\log_2$ transformed, and then subtracted from the mean $\log_2$(intensity) value of all proteins. Proteins missing in their datasets were set as $\log_2$(intensity) = 0 during the analysis. The normalization was performed for each biologically independent dataset separately. Mean normalized intensities of "kidney-only" proteome, "spleen-only" proteome as well as proteins grouped by GO terms were therefore calculated. For heatmap analysis, the data were further $z$ scored for visual interpretation. GraphPad Prism 8.0 software was used to create the graphs (box-whisker plots and heatmap).

## Comparative mitochondrial proteomics for diabetic and healthy mice

**Processing of living tissue samples.** Living kidneys from 10 to 11-week male obese-diabetic mice (db/db) and nondiabetic mice (m/m) were excised and processed as described in "CAT-S mitochondrial proteomics for living tissues". The processed cell sample from each mouse was adjusted to a volume of 15 mL with RPMI-1640.

## CAT-S mitochondrial proteomics

**Photocatalytic labeling.** Tem microliter processed samples from db/db mice and m/m mice were used for comparative proteomics. Photocatalytic labeling was performed according to the same protocol for the kidney (described in "CAT-S mitochondrial proteomics for living tissues"), using 100 nM **Ir1** photocatalyst and 100 μM **SF2** probe. A negative control (Ctrl) sample for further background subtraction was made by omitting the **Ir1** photocatalyst during the labeling using a 1:1 mixture of 5 mL processed samples from db/db and m/m mice.

**MS sample preparation.** Cell lysis, protein enrichment, and enzymatic digestion processes were all performed according to the general protocol described in "CAT-S mitochondrial proteomics for general cells". For dimethyl labeling, the three samples (diabetic group, nondiabetic group, and negative control) were subject to triplex stable isotopic labeling to yield heavy ($^{13}CD_2O$ and $NaBD_3CN$, for diabetic group), medium ($CD_2O$ and $NaBH_3CN$, for nondiabetic group) and light ($CH_2O$ and $NaBH_3CN$, for (−) catalyst control)-labeled samples, which were combined at a ratio of 1:1:1 (v/v/v) to generate a triplex dimethyl-labeled sample. The sample was further desalted on C18 tips (Thermo, 87784) according to the manufacturer's protocol and evaporated to dryness in a vacuum concentrator.

**LC–MS/MS acquisition.** The triplex sample was analyzed on Orbitrap Fusion Lumos LC–MS (Thermo) according to the general protocol (described in "CAT-S mitochondrial proteomics for general cells"), using a 180-min gradient method (0 min 2% B; 8 min 2% B; 9 min 10% B; 183 min 44% B; 188 min 99% B; 198 min 99% B; and 199 min 2% B; flow rate, 300 nL/min) to enhance the performance of data acquisition.

**Data analysis.** The obtained MS data were interpreted using Max-Quant v1.6.10 software searched against a mouse UniProt database containing 17,107 proteins. The minimum peptide length was set to seven. Half-tryptic termini and up to two missing trypsin cleavages are allowed. The quantification of heavy/light, medium/light, or heavy/medium ratios was calculated with a mass tolerance of 20 ppm. Carbamidomethylation at cysteine (+57.0215 Da) and isotopic

modifications (+28.0313, +32.0564, and +36.0757 Da for light, medium, and heavy labeling, respectively) at lysine N-terminal were set as fixed modifications. Oxidation at methionine (+15.9949 Da) and acetylation of N-terminal (+42.0106 Da) were set as variable modifications. Contaminants and proteins identified as reverse hits were removed. Proteins detected with unique peptides <2 were also removed.

For each biologically independent experiment, cutoff analysis was performed for diabetic and nondiabetic protein lists to filter off background noise (proteins with a ratio lower than the threshold were removed), based on the heavy/light (diabetic/Ctrl) and medium/light (nondiabetic/Ctrl) ratios, respectively. To maximize the retention of captured mitochondrial proteins, a less stringent cutoff ratio set to 1.2 was applied to remove obvious non-biotinylated proteins (i.e., Filter#1), yielding the "biotinylated" protein lists. Then, proteins detected in both diabetic and nondiabetic "biotinylated" lists were retained, while the others were removed (i.e., Filter#2). Lists of three biologically independent experiments were further intersected to generate the final proteomic list. Volcano plot was generated by using GraphPad Prism 8, based on the mean heavy/medium ratios and *p* values. GO enrichment analysis was performed by Metascpe (https://metascape.org). PPI network analysis was performed by STRING, with only high-confidence (score > 0.7) PPIs displayed.

**Immunoblotting analysis of protein candidates**. Whole-cell lysates and mitochondrial lysates were taken as small fractions from the samples during MS sample preparation steps, measured by BCA quantification, and mixed with 5× SDS-PAGE loading buffer to a final concentration of 1×. Samples of nondiabetic and diabetic mice were adjusted to the same protein concentration using 1× SDS-PAGE loading buffer based on the BCA quantification. Immunoblotting was performed with 4–15% gradient Tris-glycine SDS-PAGE gel and 0.20 μm PVDF membrane. Rabbit anti-CPT1B pAb (Abcam, ab134988, 1:1,000 dilution), rabbit anti-ACSM2A (Abcam, ab181865, 1:2,000 dilution), and rabbit anti-Aldh3a2 (Abcam, ab184171, 1:2,000 dilution) were used as primary antibodies while HRP-linked anti-rabbit IgG (Cell Signaling Technology, 7074 S, 1:5,000 dilution) was used as the secondary antibody.

**Mitochondrial proteomic profiling for human primary T cells**
**Isolation of primary T cells from human PBMCs**. Primary T cells were isolated from human PBMCs using EasySep™ human T cell isolation kit (STEMCELL technologies, 17951) according to the manufacturer's protocol. In our cases, approximately $4.6 \times 10^7$ human T cells with 99% purity (CD3-positive) were isolated per $10^8$ PBMCs.

**CAT-S mitochondrial proteomics**
**Photocatalytic labeling**. Approximately $3 \times 10^7$ freshly isolated primary T cells were used for each labeling group (Note: since human primary T lymphocytes are small cells with 8–10 μm diameter, each labeling group contained <0.5 mg total cellular protein). Photocatalytic labeling was performed according to the general protocol for suspension cells (described in "CAT-S mitochondrial proteomics for general cells") with the following optimizations: (1) all the incubation steps and irradiation step were performed in a 6-cm dish with 5 mL RPMI-1640 due to the smaller sample size; (2) **Ir1** concentration was optimized to 20 nM to reduce background labeling noise. A group using equivalent cells without **Ir1** photocatalyst was used as a negative control.

**MS sample preparation**. Cell lysis, protein enrichment, enzymatic digestion, and isotopic labeling processes were all performed according to the general protocol described in "CAT-S mitochondrial proteomics for general cells".

**LC–MS/MS acquisition**. Samples were all run under the same condition (120-min gradient method on Orbitrap Fusion Lumos LC–MS) described in "CAT-S mitochondrial proteomics for general cells".

**Data analysis**. The obtained MS data were interpreted using MaxQuant v1.6.10 software as described in "CAT-S mitochondrial proteomics for general cells", searched against a human UniProt database containing 20,378 proteins. Contaminants and proteins identified as reverse hits were removed. Proteins detected with unique peptides <2 were also removed. For each donor's sample, cutoff analysis based on the averaged heavy/light intensity ratio was further performed to filter off background noise (proteins with a ratio lower than the threshold were removed). Proteins detected only in the (+) catalyst group were retained. Taking mitochondrial specificity as the paramount criterion, the cutoff ratio was set to 7.0 for all donor's samples based on the distribution profiles of proteins with and without mitochondrial annotation according to MitoCarta3.0 or UniProt (2022/10/13) database, yielding a final proteome with specificity around 60%.

## Statistics and reproducibility
Statistical analysis methods of the data were detailed in the subsections above. All CAT-S proteomic experiments were performed in three biological replicates. All other experiments based on cells or animal samples were performed in ≥2 biological replicates (with replicate number *n* denoted in figure legends). No statistical method was used to predetermine the sample size. All samples and cells were randomly allocated into experimental groups. The investigator was blinded when grouping diabetic and nondiabetic mice, and blinded to group allocation when imaging "mito orphans" and controls. For other experiments, the investigators were not blinded to sample identity, since the data was from objective quantitative methods so subjective bias was not relevant. No data were excluded from the analyzes.

## Reporting summary
Further information on research design is available in the Nature Portfolio Reporting Summary linked to this article.

## Data availability
Data supporting the findings of this study are available in the Article and Supplementary Information. Source data are provided with this paper. The mass spectrometry proteomics data generated in this study have been deposited to the ProteomeXchange Consortium via the PRIDE partner repository with the dataset identifier PXD045791. Protein annotation information was obtained from UniProt (https://www.uniprot.org) and MitoCarta (https://www.broadinstitute.org/mitocarta) databases. Biological function and pathway information was obtained from the Gene Ontology database (https://geneontology.org). Protein–protein interaction information was obtained from the STRING database (https://cn.string-db.org). Previously published mouse tissue proteomics datasets for cross-analysis are available on ProteomeXchange via identifier PXD030062. Source data are provided with this paper.

## Code availability
The codes[62] used in the data analysis are available in the Supplementary Software file and on GitHub at https://github.com/hefei8alex/CAT-S_scripts (DOI identifier: https://doi.org/10.5281/zenodo.10720280).

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

## Acknowledgements

We acknowledge funding from the National Natural Science Foundation of China (22222701, 22077004, 22321005, 92253301 to X.F., 21937001, 22137001 to P.R.C.), the Ministry of Science and Technology (2022YFA1304700, 2019YFA0904201 to X.F., 2022YFE0114900 to P.R.C.), Beijing Municipal Science and Technology Commission Project (Z200010, Z221100007422046), Beijing Hospitals Authority Clinical Medicine Development Funding (YGLX202338) and Li Ge-Zhao Ning Life Science Junior Research Fellowship to X.F. Dr. Yicheng Weng is acknowledged for helpful discussion during the project. Ziyue Su and Xinyue Chen are acknowledged for their assistance in experiment design and material preparation.

## Author contributions

X.F. and P.R.C. conceived and supervised the project. Z.L., F.G., Y.Z., S.Q., and Y.H. performed the experiments and analyzed the data. F.L. helped with the mice and human blood experiments. H.G. helped with material preparation and data analysis. Z.L., X.F., and P.R.C. wrote the paper with input from all authors.

## Competing interests

X.F., P.R.C., and Z.L. have filed a patent application to the China National Intellectual Property Administration pertaining to the CAT-S method of this work (application number 202310128268.2). The remaining authors declare no competing interests.
