## [Peer Review File · Nature Communications]

REVIEWER COMMENTS

Reviewer #1 (Remarks to the Author):

In this study, Liu et al report the use of an iridium photocatalyst together with a novel probe, thio-quinone methide, for mitochondrial proteomics in a variety of cell types. The method builds upon their 2021 JACS paper (<https://pubs.acs.org/doi/10.1021/jacs.1c09171>) in which they reported the use of an iridium photocatalyst with a quinone methide (lacking the thio group) for mitochondrial proteomics. This method (in the 2021 paper and the current manuscript) relies on the passive diffusion of the iridium photocatalyst into mitochondria, followed by photocatalytic uncaging of the quinone methide, which tags nucleophilic amino acids on nearby proteins. In the current study, the authors have extended their method by adding a sulfur atom to the probe, which based on previous literature is expected to make the thio-QM probe more reactive at neutral pH. Consistent with their expectation, the authors found that the thio-QM probe gave stronger labeling of proteins, both in vitro and in living mammalian cell lines. The authors tested a panel of 6 Ir photocatalysts and identified one that gave the strongest labeling in a mammalian cell line. Impressively, the authors go on to apply their Ir photocatalyst / substrate system (“Ir1” / “SF2”) for mitochondrial proteomics in a variety of mammalian cell types, including cells dissociated from mouse tissues and T cells extracted from human patients. The current study yielded several insights into mitochondrial biology, including the identification of multiple previously unrecognized mitochondrial proteins, the upregulation of certain mitochondrial proteins in the mouse kidney and spleen, and alterations in mitochondrial protein levels related to lipid metabolism in diabetic mice.

Overall, this is a strong study featuring a novel proximity labeling probe and an impressive amount of application to diverse biological samples. However, I do have some concerns, which are described below. Assuming the authors can address these concerns in a revised manuscript, I would be supportive of publication in Nature Communications.

- In the introduction, the authors argue that the thio QM intermediate should label proteins more efficiently due to its higher reactivity. Similarly, they argue in their description of the in vitro protein labeling results that the higher electrophilicity of the thio QM intermediate can explain the increased amount of protein labeling. However, if the probe is simply more electrophilic, then wouldn't water also be expected to react faster with it – making it unclear whether we would actually expect higher yield of protein labeling? The authors should try to find literature precedent supporting the idea that the thio QM probe will show enhanced reactivity with nucleophilic amino acids on the surface of proteins, relative to the rate of reaction with water. Maybe some amino acids are softer nucleophiles than water and hence they may react faster with the thio QM intermediate?
- In line 168, the authors state that they are seeing better mitochondrial specificity than enzymatic methods. They cite reference 27. However, reference 27 (Branon et al) focuses on obtaining mito matrix

proteomes. Did Branon et al calculate specificity for mito matrix proteins, or did they just confirm that they were mitochondrial? Specificity should be assessed in the same way in this study as in the Branon et al study, if this type of direct comparison is to be made

- In line 181, the authors state that the data suggest their photocatalytic method tags mitochondrial proteins in an unbiased way, regardless of sub-mitochondrial localization. The authors argue elsewhere in the manuscript that they believe the trapping of the Ir photocatalyst in mitochondria is due to the mitochondrial membrane potential. Given this, would the Ir photocatalyst be expected to be exclusively in the mitochondrial matrix? If so, it seems that the tagging of all mitochondrial proteins by their thio QM probe would imply that the thio QM intermediate readily diffuses across the mitochondrial inner membrane. The authors should comment on this explicitly in the main text, as this has important implications for which contexts the method will be useful in.
- The authors write: “According to the zoom-in view, the phosphatase PTPN129 was mainly located on the mitochondrial outer membrane,...” Please clarify this. Do they believe that conventional fluorescence microscopy is showing them if it’s on the mito outer membrane? This does not seem possible without super resolution fluorescence imaging.
- The authors claim that TMEM160 was never previously reported to be a mitochondrial protein. However, from a quick online search, I found this 2022 paper that identifies this protein as mitochondrial: <https://febs.onlinelibrary.wiley.com/doi/10.1002/2211-5463.13496>. Therefore, the authors should amend the manuscript to remove the claim that this protein is a “mito orphan.” As far as I can see from a quick online search, it appears that the other mito orphans identified by the authors are indeed novel – but I urge the authors to be very thorough in their searching to avoid making erroneous claims and missing key citations.
- The authors should comment further on the mito orphans they identified. Were these the most abundant proteins in their dataset that lacked prior mitochondrial annotation? Did the authors test additional proteins for mitochondrial localization aside from these 4, and if so, what happened? Did any of these proteins fail to localize to mitochondria? If so, this is important to disclose in supplementary figures, as it has implications for whether the reactive intermediate generated in their PL method crosses the mito inner membrane.
- A fascinating consideration for this methodology with important practical implications is the question of which amino acids are tagged by the reactive intermediate. The authors state that a variety of nucleophilic amino acids are tagged, but at present, it appears the only data they provide on this is Extended Data 1, featuring one representative LC-MS/MS dataset. The authors should include additional LC-MS/MS datasets as supplementary figures, and if possible, they should provide supplementary figures and make comments in the main text about which amino acids were labeled most frequently.
- Line 135: the authors state that photocatalyst Ir1 gave the most labeling of proteins. They say it is presumably because of higher reactivity. But can they rule out higher concentration in mitochondria? They should comment on this. What about in vitro protein labeling activity to test activity in a more isolated context for the various photocatalysts?
- Methods section: model protein labeling in vitro. What is the labeling time for the non-UV experiment? Does not say in methods or caption

- In methods section “Protein digestion and dimethyl labeling” and in main text: I suggest making it clear that the ratiometric analysis involves a “minus photocatalyst” condition as the negative control

Reviewer #2 (Remarks to the Author):

In the submitted manuscript by Dr. Peng Chen, Dr. Xinyuan Fan and colleagues, the authors have introduced a novel tool for labeling the mitochondrial proteome with photo-proximity, offering utility in distinguishing between cell types (K562 vs. Hek293), different tissues (kidney vs. spleen), and various disease states (obese kidney vs. healthy kidney). Additionally, this work presents mitochondrial proteome mapping results from human donors, emphasizing its potential value in translational research. While these findings are of significant interest to mitochondrial biologists, there remain a few things in this paper that require clarification and enhancement before publication in Nature Communications.

<Major points>

1. The central theme of this paper revolves around the use of an iridium dye's chemical affinity for mitochondria, independent of genetic targeting materials. The manuscript thoroughly characterizes the successful mitochondrial targeting of this iridium dye in imaging cultured cells and provides mass spectrometry data supporting its mitochondrial localization. Nevertheless, it remains uncertain whether this iridium dye functions effectively in other animal tissues, particularly in metabolically active organs like brown fat, liver, and brain, which house abundant mitochondria and lipid molecules. Furthermore, the presence of hydrophobic lipid molecules in the plasma membrane, endoplasmic reticulum and lipid droplets may hinder the iridium dye's mitochondrial targeting efficiency due to its hydrophobic nature. In such cases, the method may be unable to identify the mitochondrial proteome but instead detect interactions with the iridium dye in these specific tissues or cell types, while genetically encodable proximity labeling methods may encounter fewer issues with catalyst targeting in such harsh situations (PMID: 37884807). Consequently, the authors should explicitly address and discuss the potential advantages and disadvantages of this method in comparison to contemporaneously developed proximity labeling methods.

2. The authors employed light illumination samples as controls, without using an Ir catalyst, to select labeled proteins in various experiments. However, it remains unclear how the cutoff values were determined for this selection. For instance, the cutoff values for $\text{Log}_2(\text{fold change, +/- catalyst})$ are 3.0 in the spleen, 0.96 in the kidney, and higher in cultured cells. If there is a logical procedure for establishing

different cutoff values for each sample, the authors should provide a clear explanation within the main text.

3. Building on the previous comment, it is evident that many mitochondrial proteins appear below the cutoff values. This raises the question of whether these proteins are labeled by the probe without iridium-dye catalysis or if their presence is due to inherent affinity for the beads, independent of the labeling event. If the latter is the case, the authors should address or comment on this issue concerning the somewhat ambiguous detection of labeled proteins compared to super-resolution proximity labeling methods (PMID: 28156110, PMID: 37478010). Furthermore, the authors should show all detected mass intensities of the proteins from replicate samples available as supporting dataset files.

4. While the CAT-S approach effectively maps the mitochondrial proteome, its sub-mitochondrial proteome resolution is less impressive. Presently, the data encompasses a mixture of proteins from the mitochondrial matrix, intermembrane space, and outer mitochondrial membrane. This ambiguity may result from difficulties in identifying labeled proteins (as mentioned in the previous comment) or the long lifetime of the quinone methide intermediate (PMID: 37235653) compared to the phenoxy radical-based labeling method (PMID: 28156110, PMID: 37884807, PMID: 23371551). The authors should engage in a discussion regarding this issue if a straightforward resolution is not readily achievable.

<Minor points or comments>

5. The term 'Recombinant protein' in Figure 4 may be somewhat unclear. Typically, 'recombinant protein' strongly implies purified proteins from microorganisms like *E. coli*. Instead, consider using the term 'overexpressed orphan protein.' It's advisable to avoid the term 'recombinant' throughout the entire text.

6. On page 7, lines 169-170, you compared the mitochondrial proteome specificity (67-75%) of the biotin ligase method (BioID/TurboID/miniTurbo) with the specificity of mito-APEX (94%) in HEK293T cells (PMID: 23371551). The high specificity of the APEX method may be attributed to the shorter lifetime of phenoxy radicals compared to biotin-AMP or quinone methide. To ensure an accurate comparison with previous work, the authors should mention the results obtained with APEX in this context.

7. Regarding the tissue data, it is important to acknowledge that cells dissociated from tissues during the light illumination reaction may experience stress under artificial conditions, potentially altering the mitochondrial proteome. The authors should address this issue in the Discussion section.

8. The detection of EBP among the top labeled mitochondrial proteins in the Kidney and Spleen is intriguing. While the authors have focused on other mito-orphan proteins in the current manuscript, it might be valuable to investigate the localization of EBP within the mitochondria in Kidney and Spleen. Given its relevance to cholesterol metabolism, scientists in the field of metabolism may find the protein's mitochondrial localization highly interesting.

9. Adding the gene name of the newly characterized mitochondrial protein in this study to the abstract is recommended. Additionally, for these candidates, providing a thorough description of existing reports related to diabetes or kidney function and their relevance to organ metabolism would be beneficial for readers in the biomedical field.

Reviewer #3 (Remarks to the Author):

In this manuscript the authors introduce CAT-S, a method which improves on their previously developed method, CAT-Prox, which has allowed this method to be extended to clinically relevant but less tractable cellular models such as primary cell samples. The original technology, CAT-Prox, was a novel photocatalytic proximity labeling method that extends the use of this emerging technology to intracellular domains, specifically the mitochondria. This new method, CAT-S, incorporates the use of newly designed probes that allow for higher labeling efficiencies to achieve the labeling yield necessary to investigate primary samples, which the authors go on to demonstrate in primary mouse splenocytes and kidney cells as well as primary human T cells.

The emergence of photocatalytic proximity labeling methods is a very exciting advent for interactomics, as they are a complementary approach that do not require genetic manipulation of your sample. The expansion of these technologies from extracellular domains to intracellular mapping is a welcome innovation that is necessary for proteomic mapping of endogenous complexes, particularly in disease relevant models such as primary cell samples.

The scope of this study is limited, both in the sense that this method can only be applied to mitochondrial mapping and that there is only an incremental improvement upon a previously described method with largely proof-of-concept applications demonstrated. Regardless, the introduction of new chemical probes and catalyst pairs, as well as the demonstrated application in new disease-relevant models are still noteworthy developments for this up-and-coming family of methodologies. Furthermore, the studies performed are methodologically sound, conclusions made are supported by the data shown, and the methods section is detailed and informative. Therefore, I recommend this study for publication after the items below are addressed:

1. For figure 2C, please denote the labeling time shown in the text, the legend, and/or the figure itself.
2. When comparing SF2 labeling to SF2-UV, the authors note that they saw similar labeling yields and thus conclude that labeling was dependent on thioQM but not the arylazide moiety. It would be helpful to clarify this conclusion in the text: Arylazides can be converted to a reactive species using blue light alone without a catalyst, therefore because little to no labeling was observed with -Ir1/+blue LED, is the assumption that the decaged thioQM alone is responsible for labeling? Also, is the free arylazide reactive after decaging of thioQM and do you observe any modifications from the arylazide alone by LC-MS/MS? If there is modification from the arylazide alone, do you anticipate this causing any complications?
3. The spectra in extended data fig 1 are not very legible, please replace with higher resolution images.
4. Please expand on the empirical reactivity of the various residues where labeling was observed, e.g. labeling efficiencies observed for each residue and how this was affected by solvent accessibility.
5. In the IF images, particularly in extended data fig 2, the blue representation for the iridium photocatalyst fluorescence is quite difficult to see, e.g. the fluorescence for Ir6, though understandably diffuse, is not at all visible. Please consider changing this channel to white instead of blue.
6. For figure 2j, please consider also including the channel for iridium catalyst fluorescence to show localization of the catalyst in this experiment for completion.
7. At some point, discussion on the effects of ambient light on labeling would be helpful, ideally with data to support. For example, how much labeling do you see if you expose cells to ambient light for the same amount of time you typically expose to blue light to initiate labeling? Is it necessary to take extra precaution to keep cells in the dark (e.g. wrapped in tin foil) outside of the labeling window?
8. In figure 3a, it looks like there is some biotinylation in the absence of the catalyst that extends beyond the endogenously biotinylated proteins. Please consider showing additional controls, e.g. -light, -SF2, for each cell line. In addition, please consider showing data post-pulldown, including a silver stain.
9. For figure 3, please consider including a metric to describe the coverage of your labeling method, i.e. of all of the known mitochondrial proteins, subdomains, what percentage do you pull down and detect using CAT-S?
10. For figure 3f, it would be nice to see how these data sets overlap with known mitochondrial proteins so we can see if most/all the shared proteins are mitochondrial, and what percentage of the additional 168 proteins for CAT-S and 29 proteins for CAT-Prox are (not) mitochondrial. This may also be helpful for 3g, but maybe just described in the text rather than denoted on the figure.
11. In figure 3i, would you not expect these catalysts are expected to localize mostly to the mito matrix, therefore you would see enrichment of those mito matrix/matrix facing MIM proteins over other subdomains? Also, this panel pairs nicely with 3d and also starts to address the question of coverage I brought up in point (9), so maybe consider rearranging where you mention this submitochondrial analysis in the text.
12. For extended data fig 3, please consider also showing high-mag IF images with mito staining to show that mitochondrial morphology and health is not perturbed after labeling. Also consider discussing a

scenario where it would be important to keep growing the cells after labeling, or where you would want to label the same cells multiple times.

13. Please consider using larger figures for 4a with higher resolution, and denoting where the mito orphans you selected are in the plot.

14. Were you able to find mitochondrial localization sequences in your orphans?

15. The resolution of light microscopy cannot allow you to confidently localize PTPN1 to the mito outer membrane, I highly encourage you to remove this statement and instead consider suggesting that you hypothesize that it may localize there due to what is known about its biology.

16. If any of the mito orphans have validated antibodies, please consider showing endogenous staining (if they are expressed endogenously in the HEK cells).

17. This may be a matter of semantics, but in my opinion the CAT-S performed in primary “tissues” are more accurately described as primary cells. Even though they are not being cultured post extraction, because the labeling is done after cells are dissociated and not in intact tissue, I would refer to these as primary cell suspensions and not tissues. Please consider rephrasing this section.

18. Please comment on why the specificity is lower in the primary cells than in your cultured cell experiments, e.g. what types of proteins are you enriching more of? Do these suggest poorer targeting to the mitochondria? Does there appear to be more sticking on non-specific binders during the pulldown? Is the labeling efficiency lower? Etc.

19. Again, please consider including a metric to describe the coverage of your labeling in the mouse splenocytes/kidney primary cells.

20. Please include comparisons to other datasets to validate the kidney-only/spleen-only proteins and associated GO analysis, e.g. global proteomics on whole cells or fractionated mitochondria, even expression data (e.g. RNA-seq), either generated in house or even publicly available datasets.

21. Please briefly describe in the text what “filter 1” and “filter 2” from fig 6b were based off of (i.e. which comparisons).

22. Why was the number of proteins from experiment 3 so much larger than experiment 1 and 2? What are all of these extra proteins detected?

23. Can you expand a bit more on the biological function of Cpt1a and why this particular protein is down regulated when the other fatty acid oxidation proteins that were differentially regulated are all upregulated?

As a general note, you may want to consider a modified streptavidin pulldown protocol – you can be much harsher with your washes and this may help reduce non-specific binding and cleaner separation of your true positives.

Point-to-point response to reviewers' comments:

Reviewer #1:

In this study, Liu et al report the use of an iridium photocatalyst together with a novel probe, thio-quinone methide, for mitochondrial proteomics in a variety of cell types. The method builds upon their 2021 JACS paper (<https://pubs.acs.org/doi/10.1021/jacs.1c09171>) in which they reported the use of an iridium photocatalyst with a quinone methide (lacking the thio group) for mitochondrial proteomics. This method (in the 2021 paper and the current manuscript) relies on the passive diffusion of the iridium photocatalyst into mitochondria, followed by photocatalytic uncaging of the quinone methide, which tags nucleophilic amino acids on nearby proteins. In the current study, the authors have extended their method by adding a sulfur atom to the probe, which based on previous literature is expected to make the thio-QM probe more reactive at neutral pH. Consistent with their expectation, the authors found that the thio-QM probe gave stronger labeling of proteins, both in vitro and in living mammalian cell lines. The authors tested a panel of 6 Ir photocatalysts and identified one that gave the strongest labeling in a mammalian cell line. Impressively, the authors go on to apply their ir photocatalyst / substrate system (“Ir1” / “SF2”) for mitochondrial proteomics in a variety of mammalian cell types, including cells dissociated from mouse tissues and T cells extracted from human patients. The current study yielded several insights into mitochondrial biology, including the identification of multiple previously unrecognized mitochondrial proteins, the upregulation of certain mitochondrial proteins in the mouse kidney and spleen, and alterations in mitochondrial protein levels related to lipid metabolism in diabetic mice.

Overall, this is a strong study featuring a novel proximity labeling probe and an impressive amount of application to diverse biological samples. However, I do have some concerns, which are described below. Assuming the authors can address these concerns in a revised manuscript, I would be supportive of publication in Nature Communications.

Response: We are grateful for the valuable comments from our reviewer.

- In the introduction, the authors argue that the thio QM intermediate should label proteins more efficiently due to its higher reactivity. Similarly, they argue in their description of the in vitro protein labeling results that the higher electrophilicity of the thio QM intermediate can explain the increased amount of protein labeling. However, if the probe is simply more electrophilic, then wouldn't water also be expected to react faster with it – making it unclear whether we would actually expect higher yield of protein labeling? The authors should try to find literature precedent supporting the idea that the thio QM probe will show enhanced reactivity with nucleophilic amino acids on the surface of proteins, relative to the rate of reaction with water. Maybe some amino acids are softer nucleophiles than water and hence they may react faster with the thio QM intermediate?

Response: We thank our reviewer for the insightful comment regarding QM chemistry and the potential competition between biomolecule and water reactions. We agree that the enhanced electrophilicity of thioQM presents a trade-off, potentially shortening the diffusion radius while decreasing protein labeling efficiency. However, as suggested by the reviewer, the soft electrophilic nature of thioQM might offer an additional advantage in this complex reaction environment (as verified by the enhanced mitochondrial specificity shown in Figure 3d). The presence of sulfur, with its larger valence orbitals and higher polarizability compared to oxygen, imbues thioQM with a softer electrophilic character, leading to preferential reactions with soft nucleophilic amino acid residues present in proteins over the harder nucleophile, water. This phenomenon aligns with the

fact that various soft electrophiles (e.g., metabolites or toxicants with α,β -unsaturated carbonyl motifs) remain stable in water but exhibit reactivity towards soft nucleophilic protein residues within cells (as reviewed in PMID: 30826385). Therefore, we have elaborated on this point in the revised manuscript, stating that: "... In addition, thioQM would also act a soft electrophile given the high polarizability of the sulfur atom, which favors reactions with soft nucleophiles (e.g., protein residues¹⁹) over aqueous medium, potentially increasing the labeling efficiency."

- In line 168, the authors state that they are seeing better mitochondrial specificity than enzymatic methods. They cite reference 27. However, reference 27 (Branon et al) focuses on obtaining mito matrix proteomes. Did Branon et al calculate specificity for mito matrix proteins, or did they just confirm that they were mitochondrial? Specificity should be assessed in the same way in this study as in the Branon et al study, if this type of direct comparison is to be made

Response: We have carefully checked the NBT paper again (PMID: 30125270), including its Methods section as well as proteomic datasets, and confirmed that the specificities (70%, 67% and 75%) of enzymes were calculated for whole mitochondria in that reference. As described in their Methods section (page 16 of the online PDF version), the authors calculated the **mitochondrial specificity** of labeled proteins by mito-matrix-located enzymes (Supplementary Table 7 in that paper), using a referring proteomic list containing **1555 known mitochondrial proteins** (Supplementary Table 4 in that paper) annotated by MitoCarta database or Gene Ontology term GO:0005739 (i.e., mitochondrion). Therefore, both our work and the NBT paper employed comparable methods for assessing mitochondrial specificity, enabling the direct comparison presented in our work.

- In line 181, the authors state that the data suggest their photocatalytic method tags mitochondrial proteins in an unbiased way, regardless of sub-mitochondrial localization. The authors argue elsewhere in the manuscript that they believe the trapping of the Ir photocatalyst in mitochondria is due to the mitochondrial membrane potential. Given this, would the Ir photocatalyst be expected to be exclusively in the mitochondrial matrix? If so, it seems that the tagging of all mitochondrial proteins by their thio QM probe would imply that the thio QM intermediate readily diffuses across the mitochondrial inner membrane. The authors should comment on this explicitly in the main text, as this has important implications for which contexts the method will be useful in.

Response: We thank our referee for raising our attention to the sub-mitochondrial analysis. We now have re-analyzed the sub-mitochondrial profile of the datasets, which reveals that CAT-S method actually prefers to label mitochondrial matrix proteins (Fig. 3i). Adopting the similar analysis to the APEX work (PMID: 23371551), we demonstrated that the percentages of matrix proteins for all the tested cell lines were obviously larger than the baseline (percentage in mitochondria database), with a matrix/(matrix + MOM + IMS) value around 0.9 for HeLa, HEK293T and K562, which suggests a preference for labeling matrix proteins.

Indeed, the CAT-S has captured a small number of mitochondrial inter-membrane space and outer membrane proteins as shown in the datasets. Aside from potential background noise, the existence of pore structures on the inner and outer membrane, such as mitochondrial permeability transition pores (mPTPs) (PMID: 9929477; PMID: 34245631), could provide another explanation for the labeling of MOM or IMS proteins.

Collectively, we have presented the analysis result in Fig. 3i, and rephrased the description for better clarity in the revised manuscript as following: “Sub-mitochondrial profiles of the captured mito-proteomes suggested the preference of CAT-S to label mitochondrial matrix proteins (Fig. 3i), in accordance with the MMP-based mitochondrial targeting ability of **Ir1**. Each of the three proteomes covered 36-37% of the entire human mitochondrial matrix proteins. The marginal capture of intermembrane space and outer membrane proteins might be attributed to extended probe diffusion facilitated by pore structures on inner membranes (e.g., permeability transition pore³²), or background noises.”

- The authors write: “According to the zoom-in view, the phosphatase PTPN129 was mainly located on the mitochondrial outer membrane, ...” Please clarify this. Do they believe that conventional fluorescence microscopy is showing them if it’s on the mito outer membrane? This does not seem possible without super resolution fluorescence imaging.

Response: Thanks for pointing out the limitations of our initial analysis of PTPN129 localization. We agree that conventional fluorescence microscopy is not enough to definitively place it on the outer membrane. We have removed this claim in the revised manuscript.

- The authors claim that TMEM160 was never previously reported to be a mitochondrial protein. However, from a quick online search, I found this 2022 paper that identifies this protein as mitochondrial: <https://febs.onlinelibrary.wiley.com/doi/10.1002/2211-5463.13496>. Therefore, the authors should amend the manuscript to remove the claim that this protein is a “mito orphan.” As far as I can see from a quick online search, it appears that the other mito orphans identified by the authors are indeed novel – but I urge the authors to be very thorough in their searching to avoid making erroneous claims and missing key citations.

Response: We thank our reviewer for bringing to our attention the recent report identifying human TMEM160 as a mitochondrial protein (PMID: 36209620). We acknowledge that our initial classification of TMEM160 as a “mito orphan” was inaccurate. Notably, our CAT-S-guided analysis independently identified TMEM160 as mitochondrial shortly before the aforementioned study’s publication, further validating the efficacy of our methodology. This concurrence strengthens the evidence for TMEM160’s mitochondrial localization and highlights the potential of CAT-S for novel protein discovery in this understudied organelle. For the other three human “mito orphans”, we have carefully searched the literature and verified that they were not previously reported as mitochondrial proteins.

We have amended the manuscript to reflect this information as following: “Additionally, we also identified TMEM160 in our analysis, and initially categorized this protein as a “mito orphan” (Supplementary Fig. 10). A recent study³⁴ published during our research period has concurrently confirmed human TMEM160 to be mitochondrial, further validating the efficacy of CAT-S-guided mitochondrial discovery.”

- The authors should comment further on the mito orphans they identified. Were these the most abundant proteins in their dataset that lacked prior mitochondrial annotation? Did the authors test additional proteins for mitochondrial localization aside from these 4, and if so, what happened? Did any of these proteins fail to localize to mitochondria? If so, this is important to disclose in

supplementary figures, as it has implications for whether the reactive intermediate generated in their PL method crosses the mito inner membrane.

Response: These identified “mito orphans” were the proteins with the highest [+/- catalyst] ratios in our dataset (Supplementary Data 1) that lacked prior mitochondrial annotation, which were observed to be close to known mitochondrial proteins during tSNE data visualization (Figure 4a). Indeed, we further tested several other proteins (DPM3, TMEM33, PRAF2, ISOC1, TMED1) with high ratios besides from these four (PTPN1, SLC35A4 uORF, TMEM160 and TRABD). While TMEM33 and TMED1 were not detectably expressed in our HEK293T system, the three other proteins (DPM3, PRAF2 and ISOC1) were not observed to be significantly colocalized with mitochondria, potentially as false positives. We have disclosed these data in Supplementary Fig. 11 in the revised Supplementary Information (SI).

- A fascinating consideration for this methodology with important practical implications is the question of which amino acids are tagged by the reactive intermediate. The authors state that a variety of nucleophilic amino acids are tagged, but at present, it appears the only data they provide on this is Extended Data 1, featuring one representative LC-MS/MS dataset. The authors should include additional LC-MS/MS datasets as supplementary figures, and if possible, they should provide supplementary figures and make comments in the main text about which amino acids were labeled most frequently.

Response: It is indeed important to investigate which protein residues are labeled by the thioQM probe. Therefore, we have listed the peptides as well as labeling sites identified by LC-MS/MS in Supplementary Table 2, and further provided a specific file containing all the associated spectra (Supplementary Data 6) in the revised SI.

In addition, we should also point out that during the careful analysis of our labeled data from the *in vitro* BSA experiment, we observed a small amount of labeling attributable to arylazide group. However, a control experiment using a thioQM probe without the arylazide group exhibited similar labeling patterns. This suggests that the thioQM moiety acts as the dominant labeling warhead. We have clarified this point in the revised manuscript as following: “Given that the arylazide group might be activated through photochemistry to generate other reactive species and side labeling²⁰, we replaced the PAB group with a routinely used protecting group ONP²¹ (removable by UV light) by synthesizing SF2^{uv} probe without arylazide moiety for comparison (Fig. 2d). Decaging of SF2^{uv} upon UV irradiation led to similar intensive labeling as the SF2 upon photocatalytic decaging, suggesting that the labeling was dominantly dependent on thioQM but not the arylazide moiety.”

- Line 135: the authors state that photocatalyst Ir1 gave the most labeling of proteins. They say it is presumably because of higher reactivity. But can they rule out higher concentration in mitochondria? They should comment on this. What about *in vitro* protein labeling activity to test activity in a more isolated context for the various photocatalysts?

Response: We thank our referee for the suggestion of *in vitro* activity test with various photocatalysts. Now we have performed this assay to compare the activities of Ir1-Ir6 (Supplementary Fig. 4), which again showed that Ir1 was the most efficient photocatalyst in accordance with Figure 2h. While the possibility of differential mitochondrial concentrations for these catalysts cannot be entirely ruled out, the *in vitro* data proves the superior activity of Ir1, which might provide clues for the catalyst performance in living cell.

- Methods section: model protein labeling in vitro. What is the labeling time for the non-UV experiment? Does not say in methods or caption

Response: We thank our referee for informing us of the issue. We previously mentioned the irradiation time (5 min) in the caption of Figure 2 but missed out in the Methods section. Now we have amended the Methods section “Model protein labeling” with this information.

- In methods section “Protein digestion and dimethyl labeling” and in main text: I suggest making it clear that the ratiometric analysis involves a “minus photocatalyst” condition as the negative control

Response: We thank our referee for the important suggestion. In the revised manuscript, we have made this information clear in the Methods sections “Protein digestion and dimethyl labeling” and “Data analysis”, as well as in the main text as following: “... A negative control was made by omitting the catalyst. After LC-MS/MS analysis for the (+) catalyst and (–) catalyst groups with thousands of proteins detected, we applied ratiometric cutoff of +/- catalyst ratios based on the distribution of true- and false-positives to filter off background signals, in order to retain denoised high-confidence proteome ...”.

Reviewer #2 (Remarks to the Author):

In the submitted manuscript by Dr. Peng Chen, Dr. Xinyuan Fan and colleagues, the authors have introduced a novel tool for labeling the mitochondrial proteome with photo-proximity, offering utility in distinguishing between cell types (K562 vs. Hek293), different tissues (kidney vs. spleen), and various disease states (obese kidney vs. healthy kidney). Additionally, this work presents mitochondrial proteome mapping results from human donors, emphasizing its potential value in translational research. While these findings are of significant interest to mitochondrial biologists, there remain a few things in this paper that require clarification and enhancement before publication in Nature Communications.

Response: We sincerely thank our referee for the appreciation of our work.

<Major points>

1. The central theme of this paper revolves around the use of an iridium dye's chemical affinity for mitochondria, independent of genetic targeting materials. The manuscript thoroughly characterizes the successful mitochondrial targeting of this iridium dye in imaging cultured cells and provides mass spectrometry data supporting its mitochondrial localization. Nevertheless, it remains uncertain whether this iridium dye functions effectively in other animal tissues, particularly in metabolically active organs like brown fat, liver, and brain, which house abundant mitochondria and lipid molecules. Furthermore, the presence of hydrophobic lipid molecules in the plasma membrane, endoplasmic reticulum and lipid droplets may hinder the iridium dye's mitochondrial targeting efficiency due to its hydrophobic nature. In such cases, the method may be unable to identify the mitochondrial proteome but instead detect interactions with the iridium dye in these specific tissues or cell types, while genetically encodable proximity labeling methods may encounter fewer issues with catalyst targeting in such harsh situations (PMID: 37884807). Consequently, the authors should

explicitly address and discuss the potential advantages and disadvantages of this method in comparison to contemporaneously developed proximity labeling methods.

Response: We agree that the hydrophobic nature of iridium photocatalyst may diminish its targeting efficiency in the environments with abundant lipids (e.g., fat tissue with lipid droplets), which is a potential disadvantage of this method. A possible strategy to counter this issue is using lipid-targeting photocatalyst as a control for denoising analysis. Indeed, genetically encoded proximity labeling methods would be more advantageous in terms of spatial specificity, given that the enzyme can be precisely and flexibly directed to target spaces by fusing with specific protein/peptide, with plentiful achievements in cell lines and transgenic animals (e.g., MAX-Tg PMID: 37884807).

In the revised manuscript, we have discussed the potential disadvantages of our method in comparison to the current proximity labeling methods, as following: “Indeed, limitations also exist for the CAT-S method in comparison to enzymatic approaches. The coverage and mitochondrial specificity were observed lower than the phenolic radical-based method APEX, probably due to the longer lifespan of QM intermediate compared to phenolic radical. The potential hydrophobic interactions (e.g., with lipid droplets in adipose samples) as well as leakage of small-molecule iridium catalyst may also compromise the spatial precision, while the genetically encoded enzymes, especially the recently emerged super-resolution proximity labeling methods⁵⁵⁻⁵⁷, could enable highly precise spatial localization and proteomic capture in cells ...”

2. The authors employed light illumination samples as controls, without using an Ir catalyst, to select labeled proteins in various experiments. However, it remains unclear how the cutoff values were determined for this selection. For instance, the cutoff values for Log₂(fold change, +/- catalyst) are 3.0 in the spleen, 0.96 in the kidney, and higher in cultured cells. If there is a logical procedure for establishing different cutoff values for each sample, the authors should provide a clear explanation within the main text.

Response: We thank our referee for pointing out the inadequacy of our description in the main text. We applied different cutoff values due to the high heterogeneity of distinct types of samples (e.g., cultured cell lines, mouse kidney, mouse spleen, primary T cells) that led to different ratio distributions. It is noted that various criteria or methods can be plausibly used to determine cutoff points, which should be applied in a context-dependent manner (PMID: 27812299). In this case, we considered that mitochondrial specificity was the paramount criterion, since it reflected the confidence level of captured mitochondrial proteome. Meanwhile, we sought to apply simple integer/half-integer values for representative cutoffs, which are commonly used in proximity labeling proteomics (PMID: 37036999; PMID: 32139536). Therefore, for the cultured cell lines (HeLa, HEK293T, K562), we applied a standard of ~80% mitochondrial specificity since it is significantly higher than the common mitochondria isolation or biotin ligase proximity labeling methods (67-75%, PMID: 30125270), yielding a representative cutoff of Fold-change (+/- catalyst) = 5.0 (rounded) for cell lines. For the more challenging primary tissues (kidney, spleen), we met the standard of >60% mitochondrial specificity since it is above the confidence level of mitochondrial proteomics for these tissues (even ultrapure mito-isolation proteomics gave only ~50% specificity for mouse kidney and 20~30% for mouse spleen. PMID: 37984987), yielding representative cutoffs of Fold-change (+/- catalyst) = 8.0 and 2.0 (rounded) for spleen and kidney respectively.

We are aware that this is important information to disclose. Now we have provided clearer explanations with details in the Methods section (“Data analysis” sub-sections), and in the main text

as following: “For cell lines: “After LC-MS/MS analysis for the (+) catalyst and (–) catalyst groups with thousands of proteins detected, we applied ratiometric cutoff of +/- catalyst ratios based on the distribution of true- and false-positives to filter off background signals, in order to retain denoised high-confidence proteome (~80% specificity) ...”

For tissues: “Due to the distinct properties between tissue types, we performed cutoff analysis based on labeling extent for kidney and spleen respectively, to obtain denoised proteomic lists of considerable confidence (>60% specificity) ...”

The distribution profiles of datasets before cutoff analysis are clearly shown in the figures now. Additionally, we have performed analysis with other common cutoff criteria (e.g., TPR-FPR, fixed specificity), which yielded protein lists with nearly identical profiles (Supplementary Fig. 7 and 12) as our presented data.

3. Building on the previous comment, it is evident that many mitochondrial proteins appear below the cutoff values. This raises the question of whether these proteins are labeled by the probe without iridium-dye catalysis or if their presence is due to inherent affinity for the beads, independent of the labeling event. If the latter is the case, the authors should address or comment on this issue concerning the somewhat ambiguous detection of labeled proteins compared to super-resolution proximity labeling methods (PMID: 28156110, PMID: 37478010). Furthermore, the authors should show all detected mass intensities of the proteins from replicate samples available as supporting dataset files.

Response: We acknowledged that a number of mitochondrial proteins were below the cutoff values, due to the fact that their labeling scores (i.e., fold-change (+/- catalyst)) were not adequately distinguishing from some non-mitochondrial background noises. As pinpointed by our referee, we reasoned that this ambiguity was mainly caused by the non-specific binding during enrichment step that increased the noises, which is a common issue faced by conventional proximity labeling (PMID: 27565350; PMID: 31955847). Potential marginal off-targeting of iridium catalyst may also reduce the resolution. For further development, we will optimize the sample preparation procedure in order to minimize the non-specific background, such as applying harsher washing steps (even a referee has kindly suggested us to use harsher washing in future works). Notably, the newly emerged super-resolution proximity labeling methods reported by Rhee et al (PMID: 28156110, PMID: 37478010) have provided a novel and efficient strategy by applying desthioBiotin probe and modification analysis, which significantly enhanced the resolution. We are also intrigued to apply this powerful strategy as future direction.

Collectively, we have discussed this issue with the comparison to super-resolution methods in the revised manuscript as following: “... The potential hydrophobic interactions (e.g., with lipid droplets in adipose samples) as well as leakage of small-molecule iridium catalyst may also compromise the spatial precision, while the genetically encoded enzymes, especially the recently emerged super-resolution proximity labeling methods⁵⁷⁻⁵⁹, could enable highly precise spatial localization and proteomic capture in cells.” “Further improvement of sample processing and analytical methods, such as optimizing enrichment workflow or applying modification-based analysis from super-resolution techniques⁵⁷⁻⁵⁹, would also enhance the resolution of CAT-S and reduce the ambiguity between target proteins and noises ...”

In addition, we have added all the intensity information of replicates in the supplementary datasets.

4. While the CAT-S approach effectively maps the mitochondrial proteome, its sub-mitochondrial proteome resolution is less impressive. Presently, the data encompasses a mixture of proteins from the mitochondrial matrix, intermembrane space, and outer mitochondrial membrane. This ambiguity may result from difficulties in identifying labeled proteins (as mentioned in the previous comment) or the long lifetime of the quinone methide intermediate (PMID: 37235653) compared to the phenoxyl radical-based labeling method (PMID: 28156110, PMID: 37884807, PMID: 23371551). The authors should engage in a discussion regarding this issue if a straightforward resolution is not readily achievable.

Response: We thank our referee for notifying the issue of sub-mitochondrial analysis. After re-analyzing the submitochondrial profiles of our datasets, we concluded that CAT-S actually had a preference to label mitochondrial matrix proteins. As shown in Figure 3i, the matrix/(matrix + MOM + IMS) values were ~0.9 for all the tested cell lines, which are obviously higher than the baseline (calculated for all mitochondrial proteins in database). Indeed, our method has captured several proteins with mitochondrial intermembrane space or outer membrane annotations, rendering CAT-S significantly more ambiguous than the phenoxyl radical-based methods. Aside from the aforementioned non-specific enrichment issue (Comment 3), we speculated that the existence of some pore structures on the mitochondrial inner membrane, such as mitochondrial permeability transition pores (mPTPs) (PMID: 9929477; PMID: 34245631) or other channels, might also facilitated the diffusion of intermediate or Ir catalyst out of matrix to label IMS and MOM proteins. In the revised manuscript, we have corrected the description of sub-mito analysis and discussed the issue in the main text as following: “Sub-mitochondrial profiles of the captured mito-proteomes suggested the preference of CAT-S to label mitochondrial matrix proteins (Fig. 3i), in accordance with the MMP-based mitochondrial targeting ability of **Ir1**. Each of the three proteomes covered 36-37% of the entire human mitochondrial matrix proteins. The marginal capture of intermembrane space and outer membrane proteins might be attributed to extended probe diffusion facilitated by pore structures on inner membranes (e.g., permeability transition pore³²), or background noises.”

<Minor points or comments>

5. The term 'Recombinant protein' in Figure 4 may be somewhat unclear. Typically, 'recombinant protein' strongly implies purified proteins from microorganisms like E. coli. Instead, consider using the term 'overexpressed orphan protein.' It's advisable to avoid the term 'recombinant' throughout the entire text.

Response: We thank our referee for the kind suggestion. Now we have changed the term “recombinant protein” to the more appropriate “overexpressed protein” in the revised manuscript.

6. On page 7, lines 169-170, you compared the mitochondrial proteome specificity (67-75%) of the biotin ligase method (BioID/TurboID/miniTurbo) with the specificity of mito-APEX (94%) in HEK293T cells (PMID: 23371551). The high specificity of the APEX method may be attributed to the shorter lifetime of phenoxyl radicals compared to biotin-AMP or quinone methide. To ensure an accurate comparison with previous work, the authors should mention the results obtained with APEX in this context.

Response: We have mentioned the results of mito-APEX for more accurate comparison in the revised manuscript as following: “...The results showed both increased specificity and mitochondrial protein number than the previously reported enzymatic studies using biotin

ligases BioID (255, 70% specificity), TurboID (209, 67% specificity) or miniTurbo (269, 75% specificity), though inferior to the highly efficient mito-APEX method (464, 94% specificity) probably due to the longer lifetime of quinone methides.”

7. Regarding the tissue data, it is important to acknowledge that cells dissociated from tissues during the light illumination reaction may experience stress under artificial conditions, potentially altering the mitochondrial proteome. The authors should address this issue in the Discussion section.

Response: We acknowledged that the labeling procedure may potentially perturb the mitochondrial proteome. It is also noted that cells after CAT-S photocatalytic labeling could retain both proliferation activity and mitochondrial membrane potential, suggesting the benign biocompatibility of this method (Extended Data Fig. 3). On the other hand, since the protein turnover time generally range from hours to days (even longer in non-dividing tissue cells) (PMID: 34800366, PMID: 29449567), the short labeling period (10-min scale) might not cause drastic change in the quantities of most mitochondrial proteins. We have discussed this issue in the revised manuscript as following: “... Although we could not exclude the potential stress brought by photocatalytic labeling on tissue cells, the overall profiles of their mitochondrial proteomes may remain unaffected since the CAT-S labeling period was short (10-min scale), in contrast to the long protein turnover time (generally from hours to days)^{37, 38} in tissue cells.”

8. The detection of EBP among the top labeled mitochondrial proteins in the Kidney and Spleen is intriguing. While the authors have focused on other mito-orphan proteins in the current manuscript, it might be valuable to investigate the localization of EBP within the mitochondria in Kidney and Spleen. Given its relevance to cholesterol metabolism, scientists in the field of metabolism may find the protein's mitochondrial localization highly interesting.

Response: We thank our referee for the inspiring suggestion. EBP (emopamil-binding protein) is a fascinating target playing roles in cholesterol metabolism while its properties have not been well-studied. In addition to our datasets of mouse tissues, we also witnessed EBP in other published datasets of mitochondrial proteomics (PMID: 35929479, PMID: 24708184). Currently, the lack of commercially available antibody for mouse EBP is the major obstacle for us to investigate its location in mouse tissues. Indirect methods including overexpression in engineered systems (e.g., transgenic mice) can be applied instead to study EBP for our future research project. Relevant works are currently ongoing in our lab.

9. Adding the gene name of the newly characterized mitochondrial protein in this study to the abstract is recommended. Additionally, for these candidates, providing a thorough description of existing reports related to diabetes or kidney function and their relevance to organ metabolism would be beneficial for readers in the biomedical field.

Response: We thank our referee for the recommendation. In the revised manuscript, we have added the gene names in the abstract, and also reinforced the description of how these altered proteins relate to kidney function, metabolism or diabetes in the main text as following: “The upregulated protein, Cpt1b, is a member of carnitine palmitoyl transferase family known for synthesizing acylcarnitine for fatty acid oxidation (FAO), and also reported to be inversely related to insulin sensitivity⁴⁶. Among the downregulated proteins, Acs2 is known for synthesizing medium-chain fatty acyl-CoA for metabolic utilization, and found with decreased expression in abnormal

kidney⁴⁷. Given these previous hints, we suspected that the reshaping of lipid metabolic network might play essential and multi-faceted roles during diabetic nephropathy. Particularly, the aberrant decrease of Aldh3a2 and Acsm2, which are enzymes responsible for converting fatty aldehydes and medium-chain/xenobiotic fatty acids respectively, would lead to the accumulation of such lipids and consequent cellular toxicities (e.g., metabolism disruption, biomolecule damage by aldehydes, signaling alterations)⁴⁸⁻⁵⁰...”

Reviewer #3 (Remarks to the Author):

In this manuscript the authors introduce CAT-S, a method which improves on their previously developed method, CAT-Prox, which has allowed this method to be extended to clinically relevant but less tractable cellular models such as primary cell samples. The original technology, CAT-Prox, was a novel photocatalytic proximity labeling method that extends the use of this emerging technology to intracellular domains, specifically the mitochondria. This new method, CAT-S, incorporates the use of newly designed probes that allow for higher labeling efficiencies to achieve the labeling yield necessary to investigate primary samples, which the authors go on to demonstrate in primary mouse splenocytes and kidney cells as well as primary human T cells.

The emergence of photocatalytic proximity labeling methods is a very exciting advent for interactomics, as they are a complementary approach that do not require genetic manipulation of your sample. The expansion of these technologies from extracellular domains to intracellular mapping is a welcome innovation that is necessary for proteomic mapping of endogenous complexes, particularly in disease relevant models such as primary cell samples.

The scope of this study is limited, both in the sense that this method can only be applied to mitochondrial mapping and that there is only an incremental improvement upon a previously described method with largely proof-of-concept applications demonstrated. Regardless, the introduction of new chemical probes and catalyst pairs, as well as the demonstrated application in new disease-relevant models are still noteworthy developments for this up-and-coming family of methodologies. Furthermore, the studies performed are methodologically sound, conclusions made are supported by the data shown, and the methods section is detailed and informative. Therefore, I recommend this study for publication after the items below are addressed:

Response: We thank our referee for the thorough review and insightful comments highlighting the development in this work. We are committed to addressing the issues and improving the manuscript.

1. For figure 2C, please denote the labeling time shown in the text, the legend, and/or the figure itself.

Response: We have denoted the labeling time (5 min) in the legend of Figure 2c in the revised manuscript.

2. When comparing SF2 labeling to SF2-UV, the authors note that they saw similar labeling yields and thus conclude that labeling was dependent on thioQM but not the arylazide moiety. It would be helpful to clarify this conclusion in the text: Arylazides can be converted to a reactive species using blue light alone without a catalyst, therefore because little to no labeling was observed with - Ir1/+blue LED, is the assumption that the decayed thioQM alone is responsible for labeling? Also,

is the free arylazide reactive after decaging of thioQM and do you observe any modifications from the arylazide alone by LC-MS/MS? If there is modification from the arylazide alone, do you anticipate this causing any complications?

Response: We thank our referee for the insightful suggestion. It was previously reported that arylazide could be activated by photochemistry to generate reactive species, which might cause labeling aside from the proposed QM pathway. To gain deeper insights, we have performed a series of investigations including labeling by the non-azide SF2-UV probe as well as LC-MS/MS identification of SF2 labeling sites, which proved that the QM structure was indeed responsible for labeling. Additionally, we also observed a few arylazide-modified sites (much fewer than SF2 modifications, see Supplementary Data 6), probably due to marginal labeling. More importantly, we further compared the corresponding arylazide probe (omitting the QM moiety) with SF2 probe for practical CAT-S labeling in living cells, which showed that the arylazide did not generate significant labeling by our system in living cells (Supplementary Fig. 3). Collectively, our results have suggested that the QM played a predominant role in the CAT-S labeling.

Now we have made clearer clarification in the revised manuscript as following: "Given that the arylazide group might be activated through photochemistry to generate other reactive species and side labeling²⁰, we replaced the PAB group with a routinely used protecting group ONP²¹ (removable by UV light) by synthesizing **SF2^w** probe without arylazide moiety for comparison (Fig. 2d). Decaging of **SF2^w** upon UV irradiation led to similar intensive labeling as the **SF2** upon photocatalytic decaging, suggesting that the labeling was dependent on thioQM but not the arylazide moiety." "We also examined the reported arylazide probe (**AzPh**)²⁴ as a non-QM control, with nearly no labeling observed in contrast to the intensive labeling by **SF2**, confirming the QM-dependent labeling by CAT-S inside living cells (Supplementary Fig. 3)."

3. The spectra in extended data fig 1 are not very legible, please replace with higher resolution images.

Response: In the revised manuscript, we have replaced the spectra in Extended Data Fig. 1 with images of higher resolution.

4. Please expand on the empirical reactivity of the various residues where labeling was observed, e.g. labeling efficiencies observed for each residue and how this was affected by solvent accessibility.

Response: According to our LC-MS/MS results, we have identified at least 8 types of potential labeled residues including Lys, Tyr, Asp, Asn, Glu, Gln, Trp and Thr, suggesting the promiscuous reactivity of QM towards nucleophilic residues as previous reported (PMID: 31644827). Among these residues, Lys (6 sites) was the most observed residue, followed by Gln (3 sites) and others (<3 sites), which might be attributed to the high nucleophilicity of amine as well as high exposure of Lys. All of these observed labeled residues were found exposed to solvent on protein surface (PDB 4F5S), while those buried nucleophilic residues were not found to be labeled.

We have expanded on this discussion in the revised manuscript as following: "Similar to the previously reported QMs²³, the **SF2**-derived thioQM warhead exhibited promiscuous reactivity to a wide range of nucleophilic protein residues (8 amino acids observed: Lys, Tyr, Asp, Asn, Glu, Gln, Trp and Thr), with Lys being labeled most frequently likely attributing to its high reactivity coupled

with surface exposure. All the detected sites were located on protein surface, while the internally buried residues were not observed to be labeled.”

5. In the IF images, particularly in extended data fig 2, the blue representation for the iridium photocatalyst fluorescence is quite difficult to see, e.g. the fluorescence for Ir6, though understandably diffuse, is not at all visible. Please consider changing this channel to white instead of blue.

Response: We thank our referee for the helpful suggestion. To enhance the contrast for the signal of iridium photocatalysts, we have changed the color of this channel to cyan which is significantly brighter (Fig. 2g and Extended Data Fig. 2). We chose cyan color instead of white, in consideration of generating explicit merged signal.

6. For figure 2j, please consider also including the channel for iridium catalyst fluorescence to show localization of the catalyst in this experiment for completion.

Response: We thank our referee for the kind suggestion. Although we hoped to observe the iridium catalyst after the CAT-S labeling, we were inaccessible to its fluorescent signal due to the following reasons: 1) the fluorescence of iridium photocatalyst is intrinsically weak (1-2 orders of magnitude weaker than the commercial dyes), making it difficult to observe at the concentration of 100 nM (CAT-S labeling condition); 2) the cell fixation, permeabilization and PBST washing steps during the IF sample preparation procedure have significantly removed the iridium catalyst from the cells.

7. At some point, discussion on the effects of ambient light on labeling would be helpful, ideally with data to support. For example, how much labeling do you see if you expose cells to ambient light for the same amount of time you typically expose to blue light to initiate labeling? Is it necessary to take extra precaution to keep cells in the dark (e.g. wrapped in tin foil) outside of the labeling window?

Response: Taking extra precautions to keep samples in the dark is indeed beneficial for our experiment, but not strictly necessary in the environment of normal laboratory. However, precautions should be taken to avoid the exposure of samples to outdoor sunlight or other strong light sources. We have performed additional experiment to investigate the effects of ambient light (Supplementary Fig. 2), which showed that normal indoor ambient light gave little to no labeling, but outdoor sunlight led to significant labeling which is comparable to blue light irradiation.

8. In figure 3a, it looks like there is some biotinylation in the absence of the catalyst that extends beyond the endogenously biotinylated proteins. Please consider showing additional controls, e.g. - light, -SF2, for each cell line. In addition, please consider showing data post-pulldown, including a silver stain.

Response: We have performed experiments for all the cell lines with additional controls (- light, -SF2) using both whole-cell and post-pulldown samples, and added the results in the revised manuscript (Figure 3a and Supplementary Fig. 6).

9. For figure 3, please consider including a metric to describe the coverage of your labeling method, i.e. of all of the known mitochondrial proteins, subdomains, what percentage do you pull down and detect using CAT-S?

Response: We have added an additional metric of matrix coverage for Figure 3 in the revised manuscript, as following: “Each of the three proteomes covered 36-37% of the entire human mitochondrial matrix proteins.”

10. For figure 3f, it would be nice to see how these data sets overlap with known mitochondrial proteins so we can see if most/all the shared proteins are mitochondrial, and what percentage of the additional 168 proteins for CAT-S and 29 proteins for CAT-Prox are (not) mitochondrial. This may also be helpful for 3g, but maybe just described in the text rather than denoted on the figure.

Response: Figure 3f and 3g represent the datasets already with mitochondrial annotation (non-mitochondrial proteins were not included). We have made clearer clarification in both the figure and caption in the revised manuscript.

11. In figure 3i, would you not expect these catalysts are expected to localize mostly to the mito matrix, therefore you would see enrichment of those mito matrix/matrix facing MIM proteins over other subdomains? Also, this panel pairs nicely with 3d and also starts to address the question of coverage I brought up in point (9), so maybe consider rearranging where you mention this submitochondrial analysis in the text.

Response: We thank our referee for the insightful comment on the issue of our submitochondrial analysis. After careful re-analyzing the submitochondrial profiles of our captured proteins, we concluded that CAT-S actually has a preference to capture mitochondrial matrix proteins (Figure 3i), given that the matrix/(matrix + MOM + IMS) values were ~0.9 for all the tested cell lines, which were obviously higher than the baseline (calculated for all mitochondrial proteins in database). This could be explained by the matrix-targeting effect of iridium photocatalyst driven by mitochondrial membrane potential. Indeed, CAT-S method has captured a small number of proteins with MOM or IMS annotations, rendering the method more ambiguous than enzymatic approaches. We reasoned that this might be attributed to the diffusive labeling of the QM probe. It is also noted that some pore structures exist on the mitochondrial inner membrane, such as mitochondrial permeability transition pores (mPTPs) (PMID: 9929477; PMID: 34245631) or solute channels, which might further enable the diffusion of reactive intermediate out of matrix to label some IMS and MOM proteins.

In the revised manuscript, we have amended the analysis in Figure 3 and the discussion in main text as following: “Sub-mitochondrial profiles of the captured mito-proteomes suggested the preference of CAT-S to label mitochondrial matrix proteins (Fig. 3i), in accordance with the MMP-based mitochondrial targeting ability of **Ir1**. Each of the three proteomes covered 36-37% of the entire human mitochondrial matrix proteins. The marginal capture of intermembrane space and outer membrane proteins might be attributed to extended probe diffusion facilitated by pore structures on inner membranes (e.g., permeability transition pore³²), or background noises.”

12. For extended data fig 3, please consider also showing high-mag IF images with mito staining to show that mitochondrial morphology and health is not perturbed after labeling. Also consider discussing a scenario where it would be important to keep growing the cells after labeling, or where you would want to label the same cells multiple times.

Response: We are grateful to our referee for the helpful suggestions. To evaluate the health of mitochondria, we envision that the detection of mitochondrial membrane potential (MMP) (e.g.,

MMP staining followed by flowcytometry analysis), which correlates with the normal function of mitochondria, might be a more statistical and quantifiable way. Now we have added the results of this assay in Extended Data Fig. 3c, showing that the MMP was largely maintained after CAT-S labeling procedure. We also envisioned that keeping the cells growing after labeling and labeling multiple times might enable the study of protein translocation between subcellular spaces. For example, we might label the mitochondrial proteins first, followed by specific biological treatment, and then capture the proteins of another subcellular space (e.g., ER, nucleus, cell surface) by either organelle isolation or an orthogonal proximity labeling method (e.g., TurboID, APEX). Analyzing the proteome after double enrichment would therefore help identify the translocated proteins.

We have added the discussion in the revised manuscript: “After performing CAT-S to label mitochondrial proteome in living cells, we collected the cells for proliferation assay and MMP measurement, to evaluate cellular and mitochondrial health respectively ... Accordingly, the cells largely maintained the MMP after CAT-S labeling, suggesting that the integrity and activity of mitochondria were also preserved (Extended Data Fig. 3c and Supplementary Fig. 9)”; “Given these advantages, we envisioned that CAT-S might further enable dynamic pulse-chase applications, such as study of protein translocation between subcellular spaces³³, which require to maintain cell growth after initial labeling followed by secondary subcellular capture process.”

13. Please consider using larger figures for 4a with higher resolution, and denoting where the mito orphans you selected are in the plot.

Response: In the revised manuscript, we have replaced Figure 4a with larger and clearer pictures with the mito-orphans denoted in the plot.

14. Were you able to find mitochondrial localization sequences in your orphans?

Response: Proteins can be localized to mitochondria via a variety of mechanisms (PMID: 19703392, PMID: 28301740), among which the pre-sequence pathway using the amphiphilic α -helix (typically 15-50 aa) at N-terminus is the most well-known. This pathway is responsible for importing many matrix/MIM protein, but relies on general physical properties instead of a particular sequence. We have investigated the sequences as well as structures (predicted or solved, from UniProt) of our orphans, and found that TMEM160 contains an obvious positively charged N-terminal α -helix, potentially as a pre-sequence. Similarly, SLC35A4 uORF also contains an N-terminal α -helix with multiple Lys, Arg and hydrophobic Leu. For PTPN1 and TRABD, such typical sequences were not found at N-terminal domain, but were found internally in the protein. We envisioned that they were located to mitochondria via other pathways independent of pre-sequence, such as carrier pathway, Cys-mediated pathway or multiple outer membrane localization pathways (PMID: 28301740), which might be intriguing targets for further biological study.

(Note: During our manuscript preparation, a paper reporting human TMEM160 as mitochondrial was published (PMID: 36217717) while we were unaware, and the database has recently added its mitochondrial annotation. Now we have removed the claim of TMEM160 as mito orphan, but instead think this could be a further strong validation of CAT-S-guided mitochondrial protein exploration)

15. The resolution of light microscopy cannot allow you to confidently localize PTPN1 to the mitochondrial outer membrane, I highly encourage you to remove this statement and instead consider suggesting that you hypothesize that it may localize there due to what is known about its biology.

Response: We thank our referee for notifying us of the issue. Common fluorescent microscopy is insufficient to resolve mitochondrial outer membrane. Now we have removed this statement in the revised manuscript.

16. If any of the mitochondrial orphans have validated antibodies, please consider showing endogenous staining (if they are expressed endogenously in the HEK cells).

Response: We appreciate the reviewer's suggestion for showing endogenous staining of the "mitochondrial orphans". We have actively searched various antibody suppliers, but unfortunately, there are currently no commercially available IF-suitable antibodies for SLC35A4, uORF, TMEM160 and TRABD. Although we have tried the antibody (Abcam, ab244207) for PTPN1, the IF signal was very weak (similar to the non-specific binding background) and vague in our tested cell lines. Although overexpression is a common method for studying protein localization (e.g., PMID: 233715), we will continue exploring options and remain hopeful for the development of validated antibodies in the future.

17. This may be a matter of semantics, but in my opinion the CAT-S performed in primary "tissues" are more accurately described as primary cells. Even though they are not being cultured post extraction, because the labeling is done after cells are dissociated and not in intact tissue, I would refer to these as primary cell suspensions and not tissues. Please consider rephrasing this section.

Response: We acknowledged that the CAT-S labeling was performed with dissociated samples. Now we have rephrased the term throughout the manuscript to avoid potential ambiguity.

18. Please comment on why the specificity is lower in the primary cells than in your cultured cell experiments, e.g. what types of proteins are you enriching more of? Do these suggest poorer targeting to the mitochondria? Does there appear to be more sticking on non-specific binders during the pulldown? Is the labeling efficiency lower? Etc.

Response: We envisioned that the lower specificity in primary cell samples might be attributed to the following reasons: 1) while the cultured cells are almost phenotypically uniform, the tissue and primary samples containing various cell types are highly heterogeneous, which might impede the labeling and enrichment process; 2) the primary cells from tissues or blood are more fragile than the cultured immortalized cells, and may also contain more dead or unhealthy cells that cause poor iridium targeting (due to impaired mitochondria) and high off-target labeling.

Now we have added a brief discussion on this issue in the revised manuscript as following:

"Across biological replicates, we finally revealed a list of 430 proteins (261 known mitochondrial proteins, 61% specificity) for mice kidney, and 357 proteins (228 known mitochondrial proteins, 64% specificity) for mice spleen (Fig. 5c-f)... The moderately lower coverage and specificity of captured proteomes from tissues than from immortalized cell lines might be attributed to the high heterogeneity and fragility of these primary samples."

19. Again, please consider including a metric to describe the coverage of your labeling in the mouse splenocytes/kidney primary cells.

Response: We have added a metric of matrix coverage as well as related discussion for mouse tissue samples in the revised manuscript as following: “The proteomes covered 20-21% of the total known mitochondrial matrix proteins of mouse ...”

20. Please include comparisons to other datasets to validate the kidney-only/spleen-only proteins and associated GO analysis, e.g. global proteomics on whole cells or fractionated mitochondria, even expression data (e.g. RNA-seq), either generated in house or even publicly available datasets.

Response: We have compared our proteomic data as well as associated GO terms with other published datasets. By analyzing the high-coverage mitochondrial proteomic data of mouse tissues by Mann group (PMID: 37984987), we found that our “kidney-only” proteome by CAT-S was more significant in their mouse kidney datasets, while our “spleen-only” proteome by CAT-S was more significant in their spleen datasets, showing overall concordance between the two works by different research groups.

In the revised manuscript, we have added the results as Supplementary Fig. 13, and briefly mentioned in the main text as following: “Cross-analysis with the reported in-depth tissue proteomic data⁴⁰, showed that our captured “kidney-only” and “spleen-only” proteomes were typically more significant in the reported dataset for the corresponding tissue (Supplementary Fig. 13), further supporting the proteomic characteristics observed by CAT-S.”

21. Please briefly describe in the text what “filter 1” and “filter 2” from fig 6b were based off of (i.e. which comparisons).

Response: We have added a brief description of “filter 1” and “filter 2” from Figure 6b in the revised manuscript as following (see Methods section for details):

“We firstly performed cutoff analysis based on \pm catalyst ratios to remove obvious non-biotinylated proteins (filter 1), and then retain the proteins detected both in diabetic and nondiabetic datasets (filter 2). Finally, we intersected the protein lists from biological triplicate to generate a high-confidence protein list (Fig. 6b), consisting of 400 proteins including 281 known mitochondrial proteins (70% specificity) ...”

22. Why was the number of proteins from experiment 3 so much larger than experiment 1 and 2? What are all of these extra proteins detected?

Response: We reasoned that the elevated protein number in exp#3 was caused by the individual variation of different mice. Unlike the phenotypically uniform cell lines, experimental variations could be much larger for individual animals. In fact, the kidney sample of diabetic mouse in exp#3 was larger than in the previous experiments (the 3 diabetic mice were weighted 40.5 g, 41.1 g and 46.6 g before execution for exp#1-3 respectively), which might affect the cellular state, dissociation and labeling processes, leading to different ratio profile and protein coverage. Additionally, given that the experiments were performed in different batches, the commonly existing batch effects across the multi-step procedures might further amplified the variation, though we have managed to minimize. We have also checked these extra proteins, which showed that 80% of them were non-mitochondrial, probably from the higher noise in exp#3. These proteins were excluded after overlapping.

23. Can you expand a bit more on the biological function of Cpt1a and why this particular protein is down regulated when the other fatty acid oxidation proteins that were differentially regulated are all upregulated?

Response: The differential regulation of Cpt1a and other fatty acid oxidation enzymes (e.g., Cpt1b, Hadha) is indeed intriguing. It is noted that the change of Cpt1a was very slight (ratio diabetic/non-diabetic = 0.94), while others of this pathway were generally upregulated. Given that Cpt1a and Cpt1b both catalyze the carnitine conjugation step, the nearly unchanged Cpt1a and significantly upregulated Cpt1b would overall contribute to the higher fatty acid oxidation capacity, in accordance with the reported phenotypes of diabetic kidney (PMID: 29456246). Interestingly, Cpt1a is mainly expressed in lipogenic tissues (e.g., liver), while Cpt1b is mainly expressed in tissues that have high fatty acid oxidation capacity (e.g., heart, skeletal muscle). We speculated that the increase of Cpt1b/Cpt1a ratio might shift the tissue to be more like fatty acid consumer, along with higher flux of fatty acid and carnitine. Similar trends could be observed for the white vs. brown adipocytes comparison and the development of rodent heart tissue (PMID: 7721804, PMID: 9355756).

We have added a brief discussion on Cpt1a in the revised manuscript as following: “Additionally, we also observed that Cpt1a, an isoform of Cpt1b that catalyzes the same reaction, stayed nearly unchanged (diabetic/non-diabetic ratio = 0.94) while Cpt1b and other enzymes related to FAO were generally upregulated. Given that Cpt1a predominates in lipogenic tissues (e.g., liver) while Cpt1b predominates in tissues with high FAO capacity (e.g., heart, skeletal muscle)⁵¹, we speculated that the increased Cpt1b/Cpt1a ratio under obese-diabetic condition might also shift the kidney to be more like fatty acid consumer, in accordance with the reported hypermetabolic phenotypes³⁵.”

As a general note, you may want to consider a modified streptavidin pulldown protocol – you can be much harsher with your washes and this may help reduce non-specific binding and cleaner separation of your true positives.

Response: We sincerely appreciate your insightful suggestion. We agree that optimizing our sample preparation protocol could be beneficial, especially by exploring harsher washes at specific stages (e.g., post-incubation washes) of the streptavidin pulldown process. We hypothesize that this could help reduce non-specific binding and potentially clarify the separation of true positives in our pulldown results, which we have observed some background noise for in our current protocol. We are definitely planning to experiment with different wash conditions in future research to evaluate their effectiveness and optimize our protocol for cleaner and more specific results.

REVIEWERS' COMMENTS

Reviewer #1 (Remarks to the Author):

The authors have done an outstanding job addressing all of my comments and questions in the revised manuscript. I read the comments from the other reviewers along with the authors' responses, and these also look excellent. I support publication of this manuscript in Nature Communications.

Reviewer #2 (Remarks to the Author):

The authors have successfully improved the manuscript during the revision process. I would like to recommend the publication of the revised manuscript in Nature Communications.

Reviewer #3 (Remarks to the Author):

Thank you for taking the time to respond to each my (and the other reviewers') comments. I have reviewed all of the textual edits and additional data included in the manuscript and have found them to sufficiently address the points raised by the reviewers; therefore I recommend this manuscript for publication without further comment.